# GABA$_A$ receptor dependent synaptic inhibition rapidly tunes KCC2 activity via the Cl$^-$-sensitive WNK1 kinase

Martin Heubl[1,2,3], Jinwei Zhang ⬤ [4,5,6], Jessica C. Pressey[1,2,3], Sana Al Awabdh[1,2,3], Marianne Renner[1,2,3], Ferran Gomez-Castro[1,2,3], Imane Moutkine[1,2,3], Emmanuel Eugène[1,2,3], Marion Russeau[1,2,3], Kristopher T. Kahle[6], Jean Christophe Poncer[1,2,3] & Sabine Lévi[1,2,3]

The K$^+$–Cl$^-$ co-transporter KCC2 (*SLC12A5*) tunes the efficacy of GABA$_A$ receptor-mediated transmission by regulating the intraneuronal chloride concentration [Cl$^-$]$_i$. KCC2 undergoes activity-dependent regulation in both physiological and pathological conditions. The regulation of KCC2 by synaptic excitation is well documented; however, whether the transporter is regulated by synaptic inhibition is unknown. Here we report a mechanism of KCC2 regulation by GABA$_A$ receptor (GABA$_A$R)-mediated transmission in mature hippocampal neurons. Enhancing GABA$_A$R-mediated inhibition confines KCC2 to the plasma membrane, while antagonizing inhibition reduces KCC2 surface expression by increasing the lateral diffusion and endocytosis of the transporter. This mechanism utilizes Cl$^-$ as an intracellular secondary messenger and is dependent on phosphorylation of KCC2 at threonines 906 and 1007 by the Cl$^-$-sensing kinase WNK1. We propose this mechanism contributes to the homeostasis of synaptic inhibition by rapidly adjusting neuronal [Cl$^-$]$_i$ to GABA$_A$R activity.

[1] Inserm UMR-S 839, 75005 Paris, France. [2] Université Pierre & Marie Curie, Sorbonne Universités, 75005 Paris, France. [3] Institut du Fer à Moulin, 75005 Paris, France. [4] MRC Protein Phosphorylation and Ubiquitylation Unit, College of Life Sciences, University of Dundee, Dundee DD1 5EH, Scotland. [5] Institute of Biomedical and Clinical Sciences, University of Exeter Medical School, Hatherly Laboratory, Exeter EX4 4PS, UK. [6] Departments of Neurosurgery, Pediatrics, and Cellular & Molecular Physiology, NIH-Yale Centers for Mendelian Genomics, Yale School of Medicine, New Haven, CT 06511, USA. Correspondence and requests for materials should be addressed to S.Lév. (email: sabine.levi@inserm.fr)

nhibitory GABAergic signaling in mature neurons depends on intracellular chloride concentration $[Cl^-]_i$. Thus, $Cl^-$ homeostasis is essential to maintain the polarity and amplitude of $Cl^-$ fluxes through the ionotropic γ-aminobutyric acid receptor (GABA$_A$R). The hyperpolarizing shift in GABA$_A$R-activated current during development depends on a functional upregulation of the neuronal $K^+$–$Cl^-$ co-transporter KCC2, which extrudes $Cl^-$ from neurons using outwardly directed $K^+$ gradient[1]. In addition to maintaining low $[Cl^-]_i$, KCC2 regulates the formation[2], function and plasticity[3, 4] of glutamatergic synapses. Consistent with its key role in regulating inhibitory and excitatory neurotransmission, alterations in KCC2 expression and function have been correlated with pathological network activity in a variety of neurological and psychiatric disorders[5–9].

We recently demonstrated that activity-dependent regulation of KCC2 membrane diffusion and clustering mediates a rapid regulation of $Cl^-$ homeostasis in neurons. Enhanced glutamatergic synaptic activity increases KCC2 membrane diffusion, leading to transporter escape from clusters located near excitatory and inhibitory synapses and endocytosis from the plasma membrane[10]. This regulation involves the $Ca^{2+}$-dependent, PP1-mediated, dephosphorylation of KCC2 at serine (S) 940 and the $Ca^{2+}$ activated, calpain-dependent cleavage of the KCC2 carboxy-terminal domain (CTD)[11, 12]. This downregulation of KCC2 may be induced upon increased glutamate release[13] or long-term potentiation of glutamatergic synapses[14] and results in a depolarizing shift of the reversal potential of GABA$_A$R-mediated currents ($E_{GABA}$)[12, 15]. GABA itself, when acting as a depolarizing and sometimes excitatory neurotransmitter during early postnatal development, also downregulates KCC2 via L-type calcium channel activation[16]. While regulation of KCC2 by synaptic excitation is well documented, it is unknown whether synaptic inhibition may regulate the membrane expression and/or activity of KCC2 in mature neurons.

Here we examined whether GABAergic inhibition modulates KCC2 membrane dynamics, clustering, stability, and function in mature hippocampal neurons. We show that GABA$_A$R activation stabilizes KCC2 at the plasma membrane. In contrast, blocking GABA$_A$R-dependent inhibition rapidly increases KCC2 membrane dynamics, reducing its membrane clustering, stability, and activity of the transporter. We show that this mechanism is mediated by chloride ions via the $Cl^-$-sensitive serine/threonine WNK1 kinase-dependent phosphorylation of KCC2 at threonines (T) 906 and 1007, two key regulatory sites of KCC2 activity during brain development[17, 18]. Together, our results reveal a novel mechanism that rapidly tunes $[Cl^-]_i$ and GABA signaling in response to acute changes in GABA$_A$R-mediated inhibitory transmission. We speculate that antagonizing WNK1 kinase activity may be a promising strategy to restore inhibition by restoring $Cl^-$ homeostasis in diseases like epilepsy, schizophrenia, and neuropathic pain.

## Results

**GABA$_A$R activation rapidly regulates KCC2 membrane diffusion**. We asked whether pharmacological modulation of GABAergic inhibition impacts the membrane dynamics of KCC2 using quantum dot-based single-particle tracking (QD-SPT) technique in cultures of hippocampal neurons (DIV 21–24). At this stage, GABA$_A$R-mediated responses are hyperpolarizing and inhibitory, as demonstrated by $E_{GABA}$ measurements using gramicidin-perforated patch-clamp and local GABA uncaging (Supplementary Fig. 1). Since changes in glutamatergic transmission affect KCC2 diffusion and surface expression[10], the impact of GABA$_A$R activity on KCC2 diffusion was explored in the presence of the $Na^+$ channel blocker tetrodotoxin TTX (1

μM), the ionotropic glutamate receptor antagonist kynurenic acid (KYN, 1 mM), and the group I/group II mGluR antagonist R,S-MCPG (500 μM). Herein, the "TTX + KYN + MCPG" condition is referred as the "control" condition (see Methods). Neurons were then acutely exposed to the GABA$_A$R agonist muscimol (10 μM) or competitive antagonist gabazine (10 μM). Whole-cell patch-clamp recordings showed that muscimol induces a persistent current that desensitizes to <50% upon 10 min of application (Supplementary Fig. 1). In contrast, gabazine led to a rapid and complete blockade of mIPSCs throughout application of the drug (Supplementary Fig. 1). Therefore, GABA$_A$R-mediated inhibition in our culture system can be rapidly stimulated or blocked upon acute bath application of muscimol or gabazine.

We examined whether activation of GABAergic inhibition by muscimol influences KCC2 diffusion using QD-SPT. Neurons were transfected at DIV 14 to express recombinant, Flag-tagged KCC2 (KCC2-Flag) and then surface labeled at DIV 22–25 with anti-Flag antibodies, specific intermediate biotinylated Fab fragments and streptavidin-coated QDs (see Methods section, and ref. [10]). Surface exploration of individual QDs was restricted to smaller areas upon muscimol exposure, as compared with control condition (Fig. 1a). Quantitative analysis performed on the bulk population (extrasynaptic + synaptic) of trajectories revealed no significant effect of muscimol on the diffusion coefficient of KCC2 (Fig. 1b). However, the slope of the mean-square displacement (MSD) vs. time function was reduced for trajectories recorded in the presence of muscimol as compared with control (Fig. 1c), indicative of increased confinement upon GABA$_A$R activation. Consistent with this observation, the median explored area (EA) was also significantly decreased upon muscimol application (Fig. 1d, Supplementary Table 1).

Next, we tested the impact of GABA$_A$R blockade on KCC2 diffusion. Both application of gabazine (10 μM) increased the surface explored by individual QDs (Fig. 1e). When examining the bulk population of QDs, gabazine increased KCC2-Flag diffusion coefficients by 1.4-fold (Fig. 1f, Supplementary Table 1). Furthermore, the MSD vs. time function displayed a steeper slope (Fig. 1g) and the median value of the explored area was increased by 1.4-fold (Fig. 1h, Supplementary Table 1) upon gabazine application, indicative of less confined trajectories. The regulation of KCC2 diffusion by gabazine was not due to a non-specific action of the drug, since a similar effect on KCC2 diffusion was induced by the GABA$_A$R channel blocker picrotoxin (100 μM) (Supplementary Fig. 2, Supplementary Table 1). We then compared the effect of gabazine on the diffusive behavior of KCC2 in extrasynaptic and synaptic domains of neurons co-transfected with KCC2-Flag and the excitatory and inhibitory synaptic markers homer1c-GFP and gephyrin-mRFP, respectively. Gabazine significantly increased KCC2 diffusion coefficient and explored area both in the extrasynaptic membrane and near excitatory and inhibitory synapses (Fig. 1i–k, Supplementary Table 1). Consistent with these observations, gabazine induced a 1.3-fold and 1.5- fold faster escape of KCC2 from the vicinity of excitatory and inhibitory synapses, respectively (Fig. 1l, Supplementary Table 1). Thus, KCC2 exhibits reduced diffusion constraints and a faster escape from synaptic domains upon GABA$_A$R blockade.

Synaptically released GABA may activate postsynaptic, as well as extrasynaptic GABA$_A$Rs through spillover. In hippocampal pyramidal cells, tonic currents carried by extrasynaptic GABA$_A$Rs show a predominant α5 subunit pharmacology[19, 20]. Tonic GABA$_A$R currents were virtually undetectable in cultured hippocampal neurons in the absence of exogenous GABA[21] (Supplementary Fig. 1). In agreement, addition of L-655,708 (50 μM), an α5-GABAAR-selective inverse agonist, had no effect on KCC2 diffusion (Fig. 2a, b, Supplementary Table 2). To check

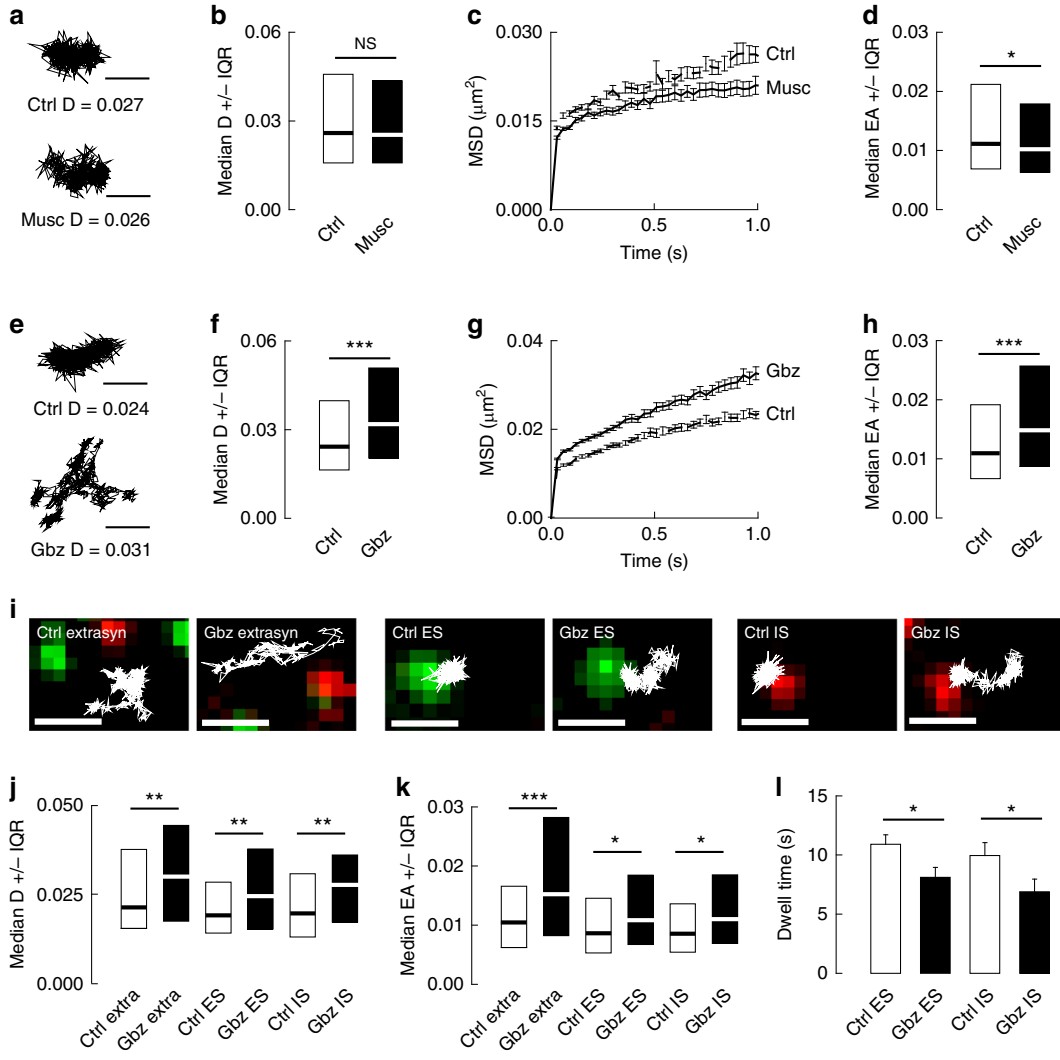

**Fig. 1** GABAAR blockade increases KCC2 membrane diffusion. **a** Examples of KCC2 trajectories showing reduced surface exploration in the presence of muscimol. Scale bars, 0.5 μm. **b** Median D ±25–75% Interquartile Range IQR of diffusion coefficients of KCC2 in control condition (white) or upon application of muscimol (black) showing no significant effect of muscimol on KCC2 diffusion. $n = 416$ QDs, 4 cultures, KS test $p = 0.139$. **c, d** Time-averaged MSD functions **c** and median explored area EA ±25–75% IQR **d** in control (white) vs. muscimol (black) conditions show increased confinement upon muscimol application. $n = 838$ QDs, 4 cultures, KS test $p = 0.039$. **e** Examples of KCC2 trajectories in the presence or absence of gabazine showing increase in QD surface exploration in gabazine-treated condition. Scale bars, 0.5 μm. **f** Median D of KCC2 in control condition (white) are increased upon gabazine application (black). $N = 441$ QDs, 5 cultures, KS test $p < 0.001$. **g, h** Time-averaged MSD functions **g** and EA **h** in control (white) vs. gabazine (black) conditions indicate decreased confinement upon GABAAR blockade. $N = 880$ QDs, 5 cultures, KS test $p < 0.001$. **i** Trajectories (white) overlaid with fluorescent clusters of recombinant homer1c-GFP (green) or gephyrin-mRFP (red) to identify extrasynaptic trajectories (left), trajectories at excitatory (middle) and inhibitory synapses (right). Scale bars, 1 μm. **j, k** Median D **j** and EA **k** of KCC2 are increased upon gabazine application (black) as compared with control condition (white). **j** extra, $n = 129$ QDs, KS test $p = 0.009$, ES, $n = 109$ QDs, KS test $p = 0.004$, IS, $n = 89$ QDs, KS test $p = 0.001$; 4 cultures. **k** extra, $n = 362$ QDs, KS test $p < 0.001$; ES, $n = 212$ QDs, KS test $p = 0.034$; IS, $n = 202$ QDs, KS test $p = 0.015$, 4 cultures. **l** Mean dwell time DT (±s.e.m) of KCC2 at excitatory (ES) and inhibitory (IS) synapses is decreased upon GABAAR blockade with gabazine (black) as compared with control condition (white). ES, Ctrl $n = 207$ QDs and Gbz $n = 218$ QDs, MW test $p = 0.035$; IS, Ctrl $n = 162$ QDs and Gbz $n = 119$ QDs, MW test $p = 0.047$, 4 cultures. **b, f, j** D in $\mu m^2 s^{-1}$; **d, h, k** EA in $\mu m^2$

the contribution of tonic GABA$_A$R currents on KCC2 diffusion, we elicited tonic GABA currents with a bath application of GABA (2 μM) 10 min before imaging. Exogenous GABA reduced by 1.2-fold KCC2 diffusion coefficients and by 1.1-fold its explored area, revealing enhanced diffusion constraints onto the transporter as compared with control (Fig. 2c, d, Supplementary Table 2). This effect is reminiscent of that induced by muscimol. Addition of L-655,708 significantly increased the mobility and reduced the confinement of KCC2, relative to GABA alone (Fig. 2c, d, Supplementary Table 2). Therefore, tonic activation of α5-

containing GABA$_A$Rs regulates KCC2 diffusion. These results indicate that both phasic and tonic GABA signaling regulate KCC2 diffusion.

GABA$_A$ and GABA$_B$ receptors (GABA$_B$Rs) both contribute to inhibitory signaling. We examined whether metabotropic GABA$_B$ receptor (GABA$_B$R)-mediated inhibition may also influence KCC2 diffusion. Bath application of the GABA$_B$R agonist baclofen (20 μM) hyperpolarized hippocampal neurons by 8.7 ± 2.2 mV (Supplementary Fig. 3). This hyperpolarization was reversible and could be blocked by the selective GABA$_B$R

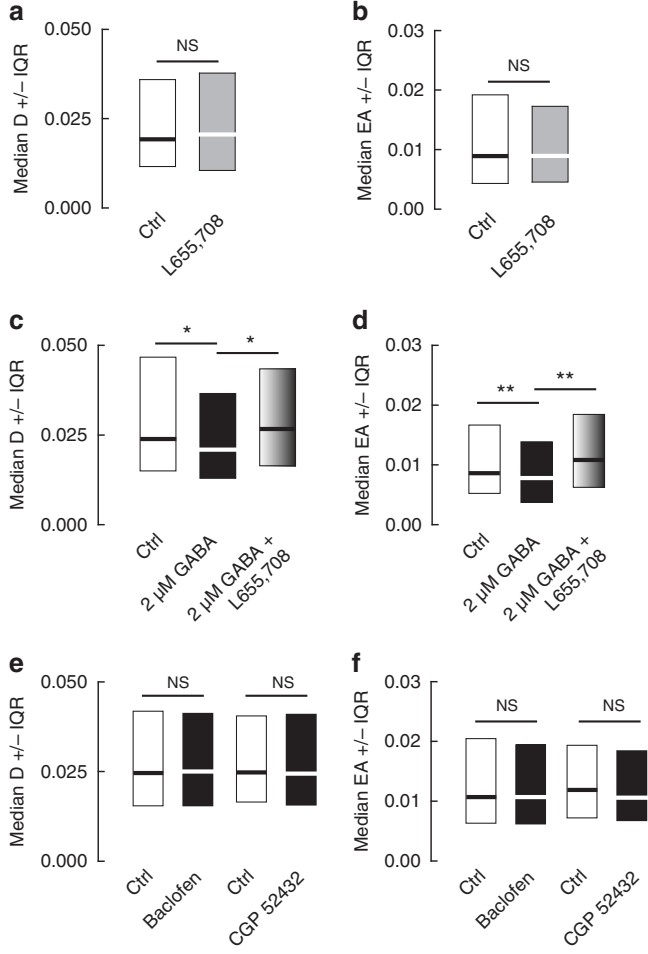

**Fig. 2** Tonic GABAAR but not GABABR activity regulates the lateral diffusion of KCC2. **a**, **b** No contribution of tonic GABAAR-mediated inhibition on KCC2 diffusion under basal conditions. Median diffusion coefficients D $\pm25$–75% IQR **a** and median explored area EA $\pm25$–75% IQR **b** (for bulk population of QDs) of KCC2 measured in absence (white) or presence (gray) of 50 µM L-655,708 in the absence of exogenous GABA. **a** $n = 320$ QDs, 2 cultures, $p = 0.658$; **b**, $n = 640$ QDs, 2 cultures, $p = 0.472$. **c**, **d** Tonic activation of GABAARs by exogenous GABA affects KCC2 diffusion. Application of 2 µM GABA (black) decreased the diffusion and increased the confinement of KCC2 as compared with control condition (white). Addition of L655,708 (pattern) to GABA decreased KCC2 diffusion constraints compared with neurons exposed to GABA only. **c** $n = 271$ QDs, 3 cultures; Ctrl vs. GABA $p = 0.042$; GABA vs. GABA + L-655,708 $p = 0.035$. **d** $n = 542$ QDs, 3 cultures; Ctrl vs. GABA, $p < 0.001$; GABA vs. GABA + L-655,708 $p < 0.001$. **e**, **f** No effect of GABABR activity on KCC2 diffusion. Median diffusion coefficients **e** and median explored area **f** (for bulk population of QDs) of KCC2 measured in absence (white) or presence (black) of baclofen or CGP52432. **e** Baclofen experiment: $n = 278$ QDs, 3 cultures, $p = 0.864$; CGP52432 experiment: $n = 279$ QDs, 3 cultures, $p = 0.425$. **f** baclofen experiment: $n = 555$ QDs, 3 cultures, $p = 0.338$; CGP52432 experiment: $n = 580$ QDs, 3 cultures, $p = 0.091$. In all graphs, KS test was used for data comparison. **a**, **c**, **e** D in µm²s-1; **b**, **d**, **f** EA in µm²

antagonist CGP52432 (20 µM, Supplementary Fig. 3). We found that up to 45 min exposure to either baclofen or CGP52432, however, had no detectable effect on diffusion coefficients and explored area (Fig. 2e, f, Supplementary Table 2). Therefore, KCC2 diffusion is specifically regulated by GABA_AR but not GABA_BR-mediated transmission. Since GABA_BR activation leads to membrane hyperpolarization, this result further suggests KCC2 diffusion may not be directly regulated by hyperpolarization.

**KCC2 modulation is independent of $Ca^{2+}$ signaling.** Rapid, activity-dependent regulation of KCC2 membrane dynamics and stability has been shown to require $Ca^{2+}$ influx through NMDAR or voltage-gated calcium channels (VGCC)[12, 15] and subsequent S940 dephosphorylation[10]. The presence of TTX and glutamate receptor antagonists in our experiments should however minimize $Ca^{2+}$ influx. In agreement, gabazine application in the presence of TTX + KYN + MCPG did not increase $[Ca^{2+}]_i$ in hippocampal neurons, as revealed in $Ca^{2+}$ imaging experiments (Fig. 3a–d). In addition, extracellular application of cadmium (100 µM; $Cd^{2+}$), a non-specific VGCC blocker, failed to prevent gabazine-induced increase in KCC2 diffusion (Fig. 3e) and reduced confinement (Fig. 3f, Supplementary Table 2).

Glutamate-induced increase in KCC2 diffusion relies on dephosphorylation of S940[10]. We asked whether phosphorylation of this residue may be also involved in the regulation of KCC2 diffusion by GABA_ARs. Overexpression of the phosphomimetic KCC2-S940D transporter[10] did not prevent the gabazine-induced increase in KCC2 diffusion (Fig. 3g, h, Supplementary Table 2). Altogether, these results demonstrate that GABA_AR-mediated regulation of KCC2 membrane diffusion does not involve $Ca^{2+}$ signaling and S940 phosphorylation.

**$Cl^-$ dependent WNK signaling regulates KCC2.** Since $Ca^{2+}$ signaling is not involved in GABA_AR-mediated regulation of KCC2 diffusion, what is the underlying signaling mechanism? GABA_AR activation leads to $Cl^-$ influx in mature neurons. Conversely, GABA_AR blockade decreases $[Cl^-]_i$. This suggests that changes in $[Cl^-]_i$ may underlie the effects of GABA_AR manipulations on KCC2 diffusion. We tested this hypothesis by first examining whether changes in GABA_AR activity were associated with changes in $[Cl^-]_i$. We measured the steady-state $[Cl^-]_i$ in neurons using fluorescence resonance energy transfer (FRET) multiphoton imaging of the chloride sensor Super-Clomeleon[22]. We calibrated the probe in our system using Neuro2a cells (see Methods section). The FRET ratio of Super-Clomeleon was tightly correlated with $[Cl^-]_I$, with highest sensitivity in the 15–100 mM range (Fig. 4a). However, the probe showed little sensitivity for concentrations ranging 0–15 mM. We analyzed the effect of muscimol or gabazine on $[Cl^-]_i$ in neurons transfected with SuperClomeleon. As expected, we observed a significant decrease in the FRET ratio of SuperClomeleon signal upon 5 min of muscimol application ($-10 \pm 3\%$; Fig. 4b), indicating an increase in $[Cl^-]_i$. Blockade of GABA_AR with gabazine is instead expected to decrease $[Cl^-]_i$. However, no detectable change in FRET ratio was observed upon 5 min. of gabazine application (Fig. 4b), likely due to the lack of sensitivity of the probe in 0–15 mM range in our experimental conditions (Fig. 4a). However, prolonged application of gabazine (50 min) led to a reduced FRET ratio of SuperClomeleon (~26 $\pm$ 8%; Fig. 4b), reflecting an increased $[Cl^-]_i$ possibly through altered transmembrane chloride extrusion (see below).

Next, we investigated whether changes in $[Cl^-]_i$ may influence KCC2 diffusion independently of GABA_AR activity. We increased $[Cl^-]_i$ by exposing neurons to the selective KCC2 inhibitor VU0463271 (10 µM)[23], while $[Cl^-]_i$ was lowered by substituting extracellular $Cl^-$ with methanesulfonate in the imaging medium. As expected for a respective increase and decrease in $[Cl^-]_i$, VU0463271 rapidly reduced the FRET ratio of SuperClomeleon signal ($-15 \pm 5\%$), while lowering extracellular $Cl^-$ level increased it ($+25 \pm 9\%$; Fig. 4b). Using these experimental paradigms, we then explored the impact of manipulating $[Cl^-]_i$ on KCC2 diffusion. Within 10 min and up to 45 min of VU application, the surface explored by individual KCC2 transporters was reduced (Fig. 4c, f) with no impact on diffusion coefficient (Fig. 4d) but a

significant increase in KCC2 confinement, as illustrated by the steeper slope of the MSD vs. time function (Fig. 4e, Supplementary Table 3). We next tested whether increasing $[Cl^-]_I$ may influence the diffusion behavior of the transporter. Using photostimulation of the light-activated chloride pump halorhodopsin (eNpHR), we found that increasing $[Cl^-]_I$ significantly reduced the diffusion coefficient and increased the confinement of KCC2. This effect was readily observed 10 s after light exposure for matched QDs (Fig. 4g) and population of QDs (Fig. 4h–j, Supplementary Table 3) and was still detected after 1 min of eNpHR activation. Therefore, increasing $[Cl^-]_I$ promotes KCC2 membrane confinement. In contrast, lowering $[Cl^-]_i$ by extracellular $Cl^-$ substitution significantly increased KCC2 surface exploration and mobility for individual trajectories (Fig. 4k) and population of QDs (Fig. 4l–n, Supplementary Table 3). Together, these results provide evidence that intracellular $Cl^-$ acts to rapidly modulate KCC2 diffusion.

The serine-threonine WNK kinases are activated by low $[Cl]_i$[24, 25] and activated WNKs promote KCC2 phosphorylation at T906 and T1007[26, 27]. Dephosphorylation of these residues correlates with increased functional expression of KCC2 during postnatal development[17, 26]. We therefore hypothesized that lowering $[Cl^-]_i$, in mature neurons may activate WNK kinase(s) and thereby promote KCC2 T906/T1007 phosphorylation and reduce its

membrane confinement. We first used quantitative polymerase chain reaction (qRT-PCR) to determine the expression pattern of the four different WNK kinase family members (WNK1–WNK4)[28] in mature hippocampal neurons. As shown in Fig. 5a, WNK1 and WNK3 mRNAs are the only WNK family members detected after 28 cycles of PCR amplification in mature hippocampal neurons. qRT-PCR revealed WNK1 is, respectively, ~500-fold, 10-fold, and 90-fold more abundant than WNK 2, 3, and 4 transcripts in DIV 21 hippocampal cultures (Fig. 5b). The relative abundance of the diverse WNK transcripts did not differ strikingly between DIV 21 hippocampal cultures and hippocampal tissue from adult rat brain (Fig. 5b). These results show that WNK1, and to a lesser extent WNK3, are the predominant WNK family members expressed in mature hippocampal neurons.

We next examined the contribution of WNK1 to the GABA_AR-dependent regulation of KCC2 diffusion. Since WNK kinases are activated by autophosphorylation of S382[27], we first tested whether GABA_AR blockade may induce WNK1 autophosphorylation. Application of gabazine increased WNK1 S382 phosphorylation 1.5-fold, indicating WNK1 activation upon GABA_AR blockade (Fig. 5c, d, Supplementary Fig. 4). WNK1 stimulates KCC2 phosphorylation via the SPAK/OSR1 kinases[27]. Consistent with WNK1 activation, gabazine application also increased SPAK S373 and OSR1 S325 phosphorylation (Fig. 5c–e, Supplementary

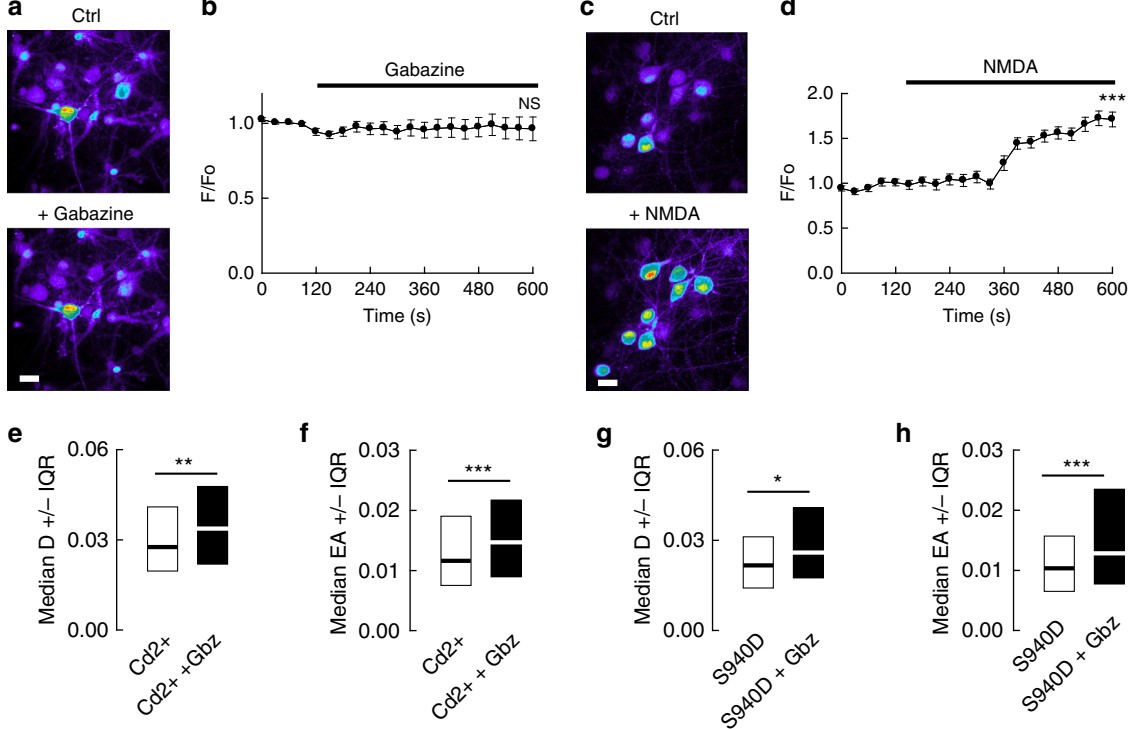

**Fig. 3** Voltage gated $Ca^{2+}$ channels and S940 dephosphorylation do not contribute to gabazine-induced increase in KCC2 diffusion. **a, b** No effect of gabazine (10 μM) on intracellular calcium level in neurons pre-treated with the Na+ channel blocker tetrodotoxin TTX (1 μM), the ionotropic glutamate receptor antagonist kynurenic acid (1 mM), and the group I/group II mGluR antagonist R,S-MCPG (500 μM). **a** Pseudocolor images of neurons loaded with Fluo4-AM, before (top) and after (bottom) gabazine treatment. Warmer colors correspond to higher Fluo4-AM fluorescence intensities. Scale bar, 10 μm. **b** Calcium level in proximal dendrites shown as F/F0 ratio (mean ± s.e.m.) measured as a function of time following gabazine application (black bar). 9 cells; 2 cultures; paired t-test $p = 0.691$. **c, d** NMDA (50 μM) increases intra-neuronal calcium level in absence of other drugs. **c** Pseudocolor images of neurons loaded with Fluo4-AM, before (top) and after (bottom) NMDA treatment. Scale bar, 10 μm. **d** Calcium level shown as F/F0 ratio as in **b**. Note the increase in intracellular calcium after NMDA exposure. 19 cells; 2 cultures; paired t test $p < 0.001$. **e, f** Median diffusion coefficients D values ±25–75% IQR **e** and median explored area EA ±25–75% IQR **f** (for bulk population of QDs) of KCC2 measured in presence of the $Ca^{2+}$ channel blocker $Cd^{2+}$ alone (white) or in presence of gabazine (black). $Cd^{2+}$ did not prevent the gabazine-induced reduction in diffusion constraints of KCC2. **e** $n = 250$ QDs, 3 cultures, $p = 0.003$. **f** $n = 500$ QDs, 3 cultures, $p < 0.001$. **g, h** Median D **g** and EA **h** (for bulk population of QDs) of KCC2-S940D under control (white) or gabazine (black) conditions. Again, S940D substitution did not prevent the gabazine-induced reduction in diffusion constraints of KCC2. **g** $n = 190$ QDs, 3 cultures, $p = 0.021$; H, $n = 380$ QDs, 3 cultures, $p < 0.001$. **e–h** KS test was used for data comparison. **e, g** D in $\mu m^2 s^{-1}$; **f, h** EA in $\mu m^2$

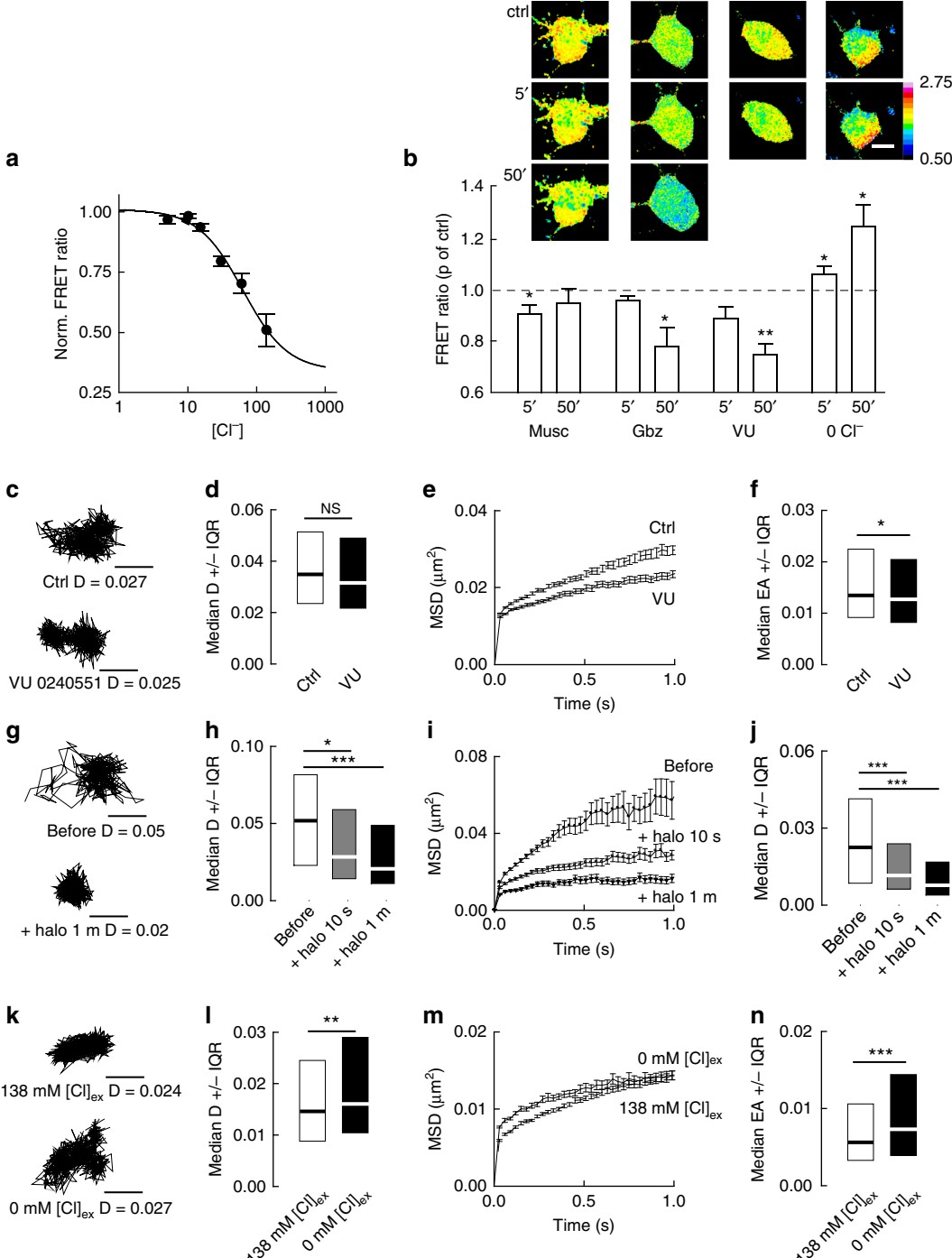

**Fig. 4** Changes in intracellular $[Cl^-]_i$ concentration tune KCC2 diffusion. **a** Calibration of SuperClomeleon in Neuro2a cells. Ionophore treatment was used to clamp $[Cl^-]_i$ to 5, 10, 15, 30, 60, and 138 mM. **b** Images of $Cl^-$ dependent changes in the FRET ratio (535/483 nm) in hippocampal neurons expressing SuperClomeleon, before (ctrl) and after 5 or 50 min application of muscimol (Musc), gabazine (Gbz), KCC2 blocker (VU) or extracellular $Cl^-$ substitution (0 Cl). The graph shows changes in FRET emission ratio upon treatment relative to control. Muscimol $n = 6$ cells, $p = 0.031$ at 5 and 0.063 at 50 min; Gabazine $n = 5$ cells, $p = 0.063$ at 5 and 0.008 at 50 min; VU $n = 4$ cells, $p = 0.125$ at 5 and 0.015 at 50 min; 0 Cl $n = 4$ cells, $p = 0.047$ at 5 and 0.030 at 50 min. Wilcoxon ranked sum test or paired $t$-test when normality test was passed; 4 cultures. Insets: Ratiometric images of SuperClomeleon in control vs. treatment. Scale bar, 10 μm. **c** KCC2 trajectories in control vs. VU. Scale bars, 0.5 μm. **d** Median diffusion coefficient D values ±25–75% IQR (for bulk population of QDs) of KCC2 in control condition (white) or upon application of VU (black). $n = 386$ QDs, 3 cultures, KS $p = 0.228$. **e, f** MSD vs. time functions **h** and median explored area EA ±25–75% IQR **i** in control (white) vs. VU (black). $N = 712$, 3 cultures, KS test $p = 0.032$. **g** Matched KCC2 trajectories before (t0), and after 10 s or 1 min of eNpHR (+halo) photostimulation. Scale bars, 0.5 μm. **h–j** Median D **h** MSD **i** and EA **j** upon 10 s or 1 min of eNpHR photostimulation. **h** $n = 215$ QDs, 2 cultures, KS test $p = 0.011$ and $p < 0.001$; **j**, $n = 469$ QDs, 2 cultures, KS test $p < 0.001$. **k** KCC2 trajectories in high vs. low extracellular $Cl^-$ concentration. Scale bars, 0.5 μm. **l–n** Median D **l** MSD **m** and EA **n** showing increased diffusion and reduced confinement of KCC2 upon reduction of $Cl^-$ concentration. **l** $n = 408$ QDs, 3 cultures, KS test $p = 0.001$; N, $n = 816$ QDs, 3 cultures, KS test $p < 0.001$. **d, h, l** D in $μm^2s$-1; **f, j, n** EA in $μm^2$

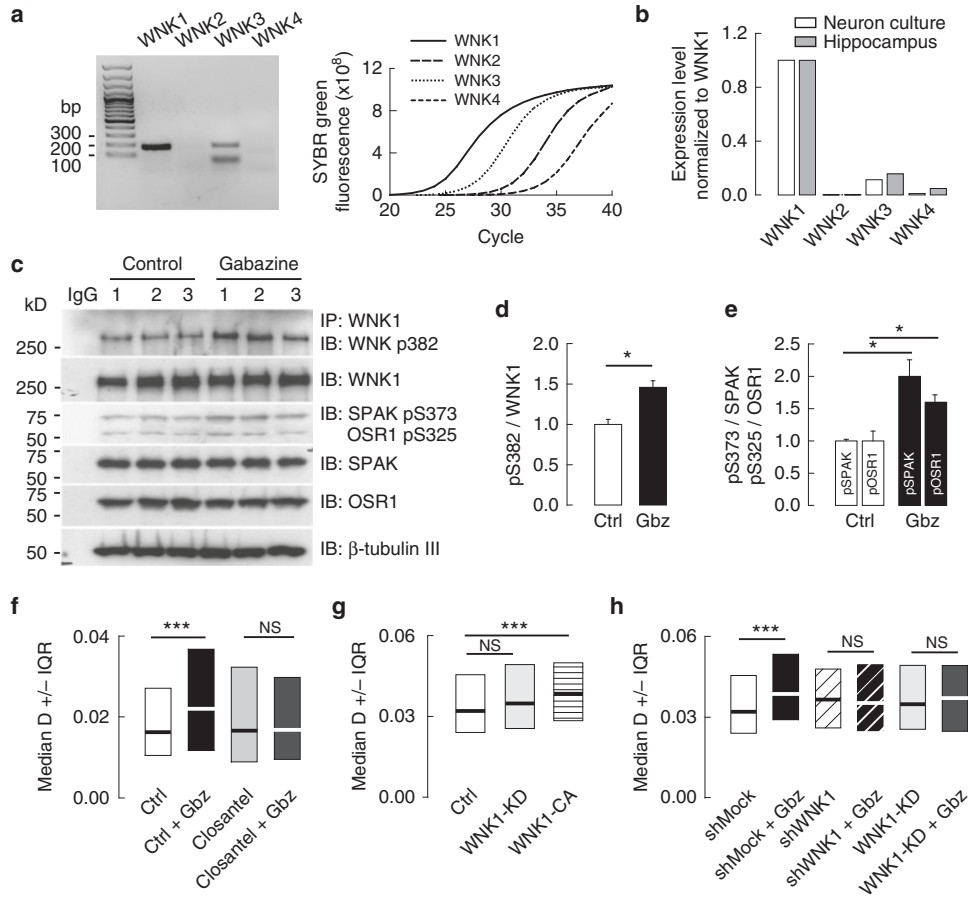

**Fig. 5** WNK1 is involved in KCC2 regulation by GABAAR-mediated transmission. **a** Left panel, Gel example showing amplification of WNK1 and WNK3 but not of WNK2 and WNK4 cDNAs in DIV 21 hippocampal neurons after 28 cycles of semi-quantitative RT-PCR. **a** Right panel, Quantitative real time polymerase chain reaction qRT-PCR of DIV 21 cultures illustrating amplification of WNK1 (plain line) followed by WNK3 (cross), WNK4 (circle) and WNK2 (dash) transcripts. **b** Results of qRT-PCR reactions for WNK1, 2, 3, and 4 from 3-week-old hippocampal cultures (white) and rat adult hippocampal tissue (gray). Expression levels of WNK2, 3 and 4 were normalized to WNK1 expression level. **c–e** Western blot and quantification (mean ± s.e.m., three independent experiments 1–3) of the activated kinases WNK1 P-S382, SPAK P-S373, and the SPAK homolog OSR1 P-S325 phosphorylation in control (white) and upon gabazine application (black), showing increased phosphorylation of WNK1 (**c**, **d** t-test p = 0.011), SPAK (**c–e** t-test p = 0.017) and OSR1 (**c–e** t-test p = 0.035) upon GABAAR blockade. **f** Median diffusion coefficient D values ±25–75% IQR (for bulk population of QDs) of KCC2 show increased KCC2 diffusion upon gabazine application is completely blocked when gabazine is applied in the presence of the SPAK/OSR1 inhibitor closantel. For each condition n = 405 QDs, 4 cultures. Ctrl (white), Gbz (black), closantel (light gray), closantel + Gbz (dark gray), Ctrl vs. Gbz p < 0.001, closantel vs. closantel + Gbz p = 0.55. **g** Median diffusion coefficients ±25–75% IQR of KCC2 (for bulk population of QDs) in neurons expressing kinase-dead WNK1 (WNK1-KD, light gray) or constitutively-active WNK1 (WNK1-CA, pattern) vs. control plasmid (white). For each condition n = 322 QDs, 5 cultures. Ctrl vs. WNK1-KD p = 0.234, Ctrl vs. WNK1-CA p < 0.001. **h** WNK1 suppression by shRNA (shWNK1) and overexpression of WNK1-KD abolished the gabazine-mediated increase in KCC2 diffusion in neurons expressing shMock. For each condition n = 322 QDs, 5 cultures. shMock (white), shMock + Gbz (black); shWNK1 (black stripe), shWNK1 + Gbz (white stripe). shMock vs. shMock + Gbz p < 0.001; shWNK1 vs. shWNK1 + Gbz p = 0.812; WNK1-KD vs. WNK1-KD + Gbz p = 0.167. In all **f–h** graphs, the KS test was used for data comparison. **f–h**, D in μm²s⁻¹

Fig. 4). Gabazine-induced SPAK S373 and OSR1 S325 phosphorylation could be inhibited by addition of the SPAK/OSR1 inhibitor closantel[29] (Supplementary Fig. 5). In contrast, and consistent with the upstream position of WNK1 in the WNK1-SPAK/OSR1-KCC2 signaling pathway, closantel had no effect on WNK1 S382 phosphorylation (Supplementary Fig. 5).

Next, we assessed the contribution of the WNK-SPAK/OSR1 signaling cascade to the GABA_AR-dependent regulation of KCC2 diffusion. Overexpressing shRNAs that reduced WNK3 expression by ~80% (Supplementary Fig. 6) did not block the effect of gabazine on KCC2 diffusion (Supplementary Fig. 6), suggesting it may primarily involve WNK1. This was tested using overexpression of kinase-dead (WNK1-KD) or constitutively-active WNK1 (WNK1-CA)[17]. Under basal conditions, overexpression of WNK1-KD did not influence KCC2 diffusion (Fig. 5g, Supplementary Table 3) whereas overexpression of a

WNK1-CA significantly enhanced KCC2 diffusion (Fig. 5g, Supplementary Table 3), consistent with previous work showing reduced membrane stability of KCC2 upon WNK1 phosphorylation[17]. Moreover, shRNA-mediated WNK1 knockdown or WNK1-KD overexpression suppressed the gabazine-induced increase in KCC2 mobility (Fig. 5h, Supplementary Table 3), as did inhibition of SPAK/OSR1 with closantel (Fig. 5f, Supplementary Table 3). Collectively, these results implicate the activity of the WNK1 and SPAK/OSR1 kinases in the GABA_AR-dependent regulation of KCC2 lateral diffusion.

WNK activity is required for KCC2 T906 and T1007 phosphorylation, which inhibits KCC2 activity[17, 26]. Consistent with an increased WNK activation upon GABA_AR blockade, gabazine application increased KCC2 T906/T1007 phosphorylation (Fig. 6a, b; Supplementary Fig. 7). Interestingly, WNK kinases not only promote KCC2 T906/T1007 but also NKCC1

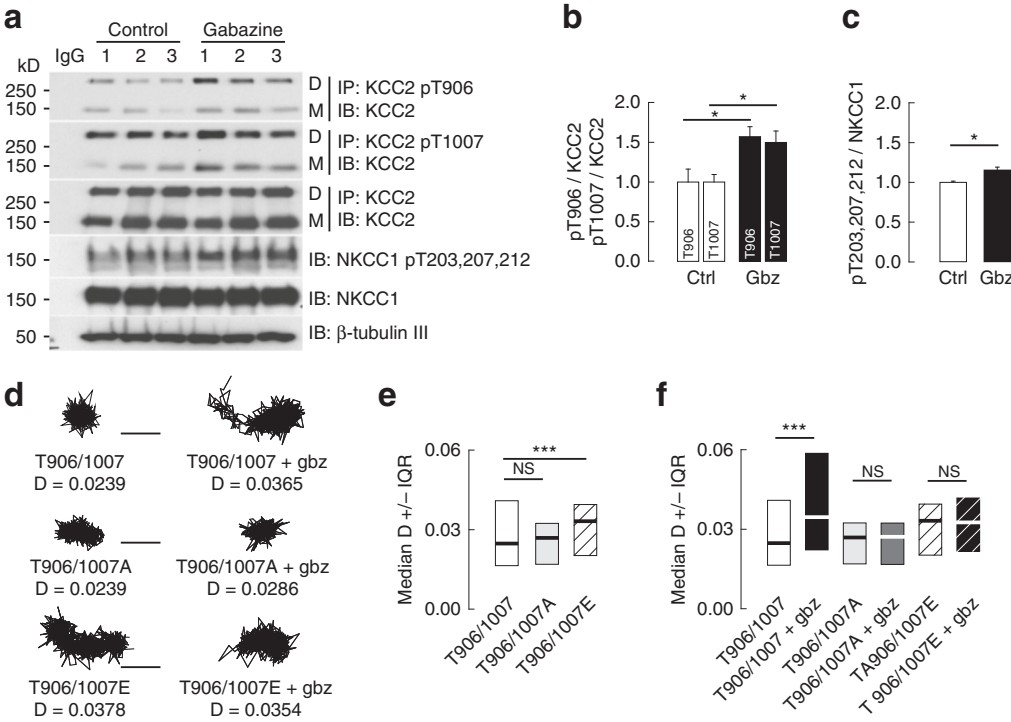

**Fig. 6** GABAAR-dependent regulation of KCC2 membrane dynamics is dependent on KCC2 phospohrylation of T906/T1007. **a–c** Western blot **a** and quantification (mean ± s.e.m., three independent experiments 1–3) of KCC2 T906/T1007 **b** and NKCC1 T203/T207/T212 **c** phosphorylation in control (white) and gabazine (black) conditions showing increased phosphorylation of KCC2 (t-test T906: p = 0.050; T1007: p = 0.046) and NKCC1 T203/T207/T212 (t-test p = 0.020) upon GABAAR blockade. **d** Examples of KCC2-T906/T1007, KCC2-T906/T1007A, and T906/T1007E trajectories in control vs. gabazine conditions. Scale bars, 0.5 μm. **e** Median QD diffusion coefficients D values ±25–75% IQR (for bulk population of QDs) of KCC2-T906/T1007 (white), KCC2-T906/T1007A (gray), and KCC2-T906/T1007E (black stripe) in resting condition show comparable D values of KCC2-T906/T1007 and KCC2-T906/T1007A transporters. In contrast, the phospho-mimetic KCC2-T906/T1007E was faster than KCC2-T906/T1007. T906/T1007 197 QDs, T906/T1007A 238 QDs, T906/T1007E 241 QDs, 3 cultures; T906/T1007 vs. T906/T1007A p = 0.932, T906/T1007 vs. T906/T1007E p < 0.001. **f** Median D (for bulk population of QDs) of KCC2-T906/T1007, KCC2-T906/T1007A, and KCC2-T906/T1007E in absence or presence of gabazine. Note that gabazine selectively reduced diffusion constraints of KCC2-T906/T1007 but not of KCC2-T906/T1007A and KCC2-T906/T1007E. T906/T1007 (white) n = 197 QDs, T906/T1007 + Gbz (black) n = 197 QDs, T906/T1007A (light gray) n = 238 QDs, T906/T1007A + Gbz (dark gray) n = 238 QDs, T906/T1007E (black stripe) n = 241 QDs, T906/T1007E + Gbz (white stripe) n = 241 QDs, 3 cultures; T906/T1007 vs. T906/T1007 + Gbz p < 0.001, T906/T1007A vs. T906/T1007A + Gbz p = 0.916, T906/T1007E vs. T906/T1007E + Gbz p = 0.648. **e, f** unless mentioned the KS test was used for all data comparison; **d** in μm²s⁻¹

T203/T207/T212 phosphorylation[30]. This in turn results in both KCC2 inhibition and NKCC1 activation[28]. Consistent with this dual modulation of Cl⁻ transport by WNK, GABAAR blockade also induced NKCC1 T203/T207/T212 phosphorylation (Fig. 6a, c). This gabazine-induced phosphorylation of KCC2 and NKCC1 was SPAK/OSR1-dependent, as it was inhibited by closantel (Supplementary Fig. 5). In addition, KCC3 which is also expressed in hippocampal neurons[31] was also phosphorylated upon gabazine application in a SPAK/OSR1-dependent manner (Supplementary Fig. 8). Finally, KCC2 may be immuno-precipitated using a phospho-specific KCC2 pT906 antibody that recognizes a phospho-residue highly homologous to other KCCs[27]. We used this antibody to control whether the detection of increased KCC2 T906 phosphorylation may be due to a contribution of other KCCs. This was not the case, since KCC1 and KCC4 were not retained after KCC2 T906 immunoprecipitation (Supplementary Fig. 8).

In order to test the involvement of KCC2 T906/T1007 phosphorylation in the gabazine-induced regulation of KCC2 diffusion, we expressed KCC2 constructs harboring mutations of T906 and T1007 to either glutamate (T906/T1007E) or alanine (T906/T1007A) that mimic phosphorylated or dephosphorylated states, respectively[17]. Under basal conditions, the diffusion of KCC2 T906/T1007E was enhanced 1.3-fold compared with WT

KCC2 (Fig. 6d, e, Supplementary Table 3). In contrast, the mobility of the KCC2 T906/T1007A did not differ significantly from that of WT KCC2 (Fig. 6d, e, Supplementary Table 3). These results indicate that the majority of membrane KCC2 in mature neurons is likely dephosphorylated on T906/T1007 under basal conditions, consistent with previous results[17, 26]. Importantly, KCC2 T906/T1007A prevented the gabazine-induced increase in KCC2 lateral diffusion (Fig. 6d, f, Supplementary Table 3). Furthermore, gabazine did not further increase the diffusion of the T906/T1007E transporter (Fig. 6d, f, Supplementary Table 3). We conclude that GABAAR-dependent regulation of KCC2 diffusion involves phosphorylation of its T906/T1007 residues.

**Functional impact of KCC2 regulation by GABAAR inhibition.** What is the functional impact of GABAAR-mediated regulation of KCC2 diffusion? We previously demonstrated that the reg-ulation of KCC2 lateral diffusion by increased excitation allows for a rapid regulation of the transporter clustering and activity[10]. We therefore examined whether changes in GABAAR-dependent inhibition also resulted in altered KCC2 clustering in hippo-campal neurons. In control conditions, KCC2-Flag formed numerous clusters along the dendrites of transfected neurons (Fig. 7a). Neither muscimol or gabazine affected the density of

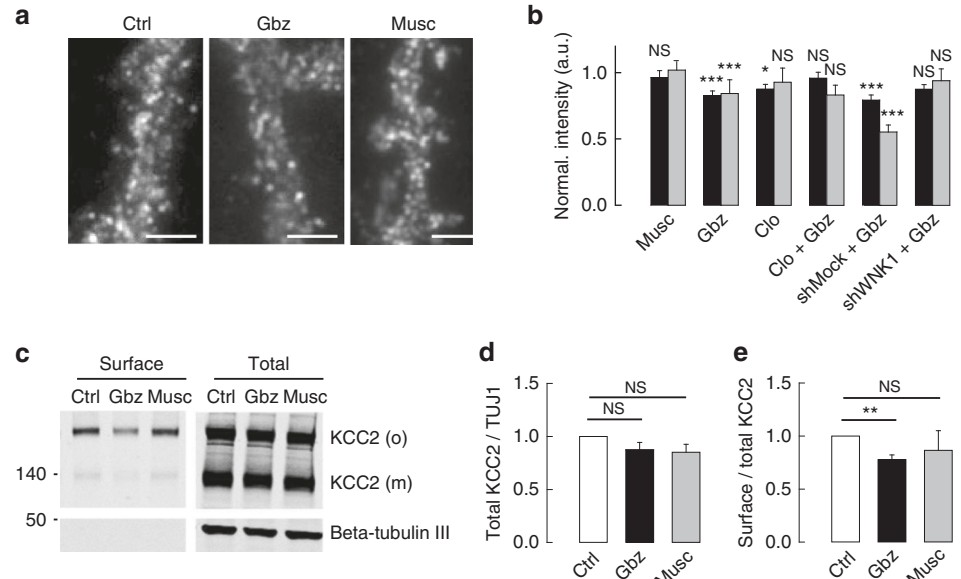

**Fig. 7** Regulation of KCC2 membrane clustering and stability by GABAAR-mediated inhibition. **a** Flag surface staining in hippocampal neurons (DIV 21–23) expressing recombinant KCC2–Flag in absence (Ctrl) or presence of gabazine (Gbz), or muscimol (Musc) for 30 min. Scale bars, 10 μm. Note the loss of KCC2 immunoreactivity after 30 min exposure to gabazine but not muscimol. **b** Quantifications of KCC2 pixel (black) and cluster (gray) intensity in each experimental condition. Note muscimol had no effect while gabazine decreased KCC2 cluster and pixel intensity. Closantel (Clo) treatment and WNK1 shRNA (shWNK1) overexpression suppressed the gabazine-induced reduction in KCC2 clustering as compared with control and shMock treated neurons, respectively. Values were normalized to the corresponding control values. The MW test was used for data comparison. Muscimol experiment: Ctrl $n = 62$ cells, Musc $n = 58$ cells, cluster intensity $p = 0.823$, pixel intensity $p = 0.470$; 4 cultures. Gabazine and closantel experiment: Ctrl $n = 77$ cells, Gbz $n = 83$ cells, cluster intensity $p = 0.006$, pixel intensity $p < 0.001$; Closantel $n = 65$ cells, cluster intensity $p = 0.05$, pixel intensity $p = 0.016$; Closantel + gabazine $n = 57$ cells, cluster intensity $p = 0.441$, pixel intensity $p = 0.448$; 4 cultures. ShWNK1 experiment: shMock, $n = 48$ cells; shMock + gabazine $n = 50$ cells, cluster intensity $p < 0.001$, pixel intensity $p < 0.001$; shWNK1 $n = 40$ cells, cluster intensity $p = 0.38$, pixel intensity $p = 0.957$; shWNK1 + gabazine $n = 53$ cells, cluster intensity $p = 0.249$, pixel intensity $p = 0.041$; 3 cultures. **c** Biotinylated surface fraction and total protein expression of KCC2 after 30 min of gabazine or muscimol treatments. **d** No change of total KCC2 protein level between control (white) and gabazine (black; $t$-test $p = 0.126$) or muscimol (gray; $t$-test $p = 0.093$) conditions (normalized to beta-tubulin III [TUJ1]). $N = 4$. **e** Quantification of the ratio of the surface pool of KCC2 over the total pool of KCC2 in control (white), gabazine (black), and muscimol (gray) conditions showing reduction of surface KCC2 after gabazine ($t$-test $p = 0.003$) but not muscimol ($t$-test $p = 0.495$) treatment ($n = 4$ experiments)

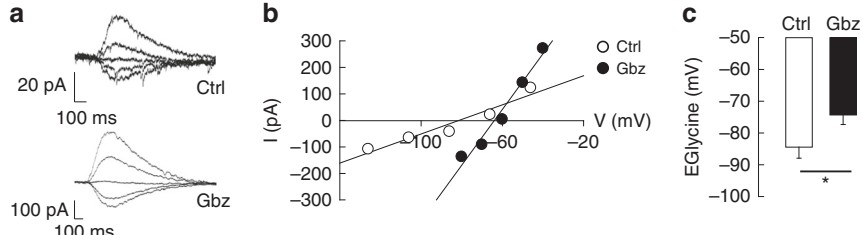

**Fig. 8** Loss of KCC2 upon GABAAR blockade correlates with increased $[Cl^-]_i$. **a, b** Typical gramicidin-perforated patch-clamp recordings of glycine receptor mediated currents induced by short (100 ms) puff of glycine at different membrane potentials in control conditions (**a** upper trace; **b** white) or upon 30 min gabazine application (**a** lower trace; **b** black). EGlycine was determined as the intercept of the I–V curve with the x-axis. **c** Gabazine application induced EGlycine depolarization ($t$-test $p = 0.038$) indicating increased $[Cl^-]_i$. Data represent the mean ± s.e.m. of 13 cells (Ctrl) and 12 cells (Gbz)

these clusters (Fig. 7a, Supplementary Fig. 9). Muscimol did not affect the mean fluorescence intensity (Fig. 7b) or the mean fluorescence intensity per pixel within clusters (Fig. 7b), indicating no detectable impact on KCC2 clustering. In contrast, gabazine application reduced membrane-associated KCC2-Flag immunoreactivity (Fig. 7a, b). A 30 min exposure to gabazine reduced by 1.2-fold the mean fluorescence intensity of clusters and the mean fluorescence intensity per pixel within clusters (Fig. 7b) as compared with untreated cells. Therefore, gabazine-induced increase in KCC2 membrane dynamics is accompanied by a rapid reduction in the clustering of the transporter. Inhibition of SPAK/OSR1 with closantel (Fig. 7b), shRNA knockdown of WNK1 (Fig. 7b), or shRNA knockdown of WNK3

(Supplementary Fig. 6) suppressed the gabazine-induced reduction in the mean fluorescence intensity of clusters and the mean fluorescence intensity per pixel within clusters. These data implicate the WNK1–WNK3/SPAK/OSR1 signaling in the gabazine-dependent regulation of KCC2 clustering.

KCC2 co-transporters that escaped clusters upon gabazine treatment may diffuse freely in the membrane or undergo clathrin-dependent endocytosis. Surface biotinylation experiments revealed that gabazine reduced cell surface KCC2 to 78 ± 9% of control without significantly changing the total protein level of the transporter (Fig. 7c–e, Supplementary Fig. 10). In contrast, muscimol did not modify KCC2 surface and total protein level (Fig. 7c–e, Supplementary Fig. 10), indicating it does

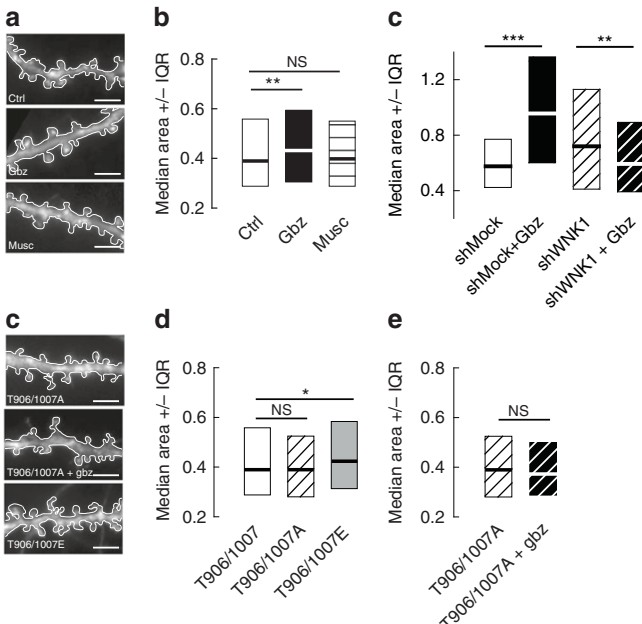

**Fig. 9** KCC2-dependent spine volume regulation is modulated by GABAAR-mediated inhibition. **a** Secondary dendrites of eGFP expressing neurons in control conditions or upon application of gabazine or muscimol. Scale bars, 10 µm. **b** Median values ±25–75% IQR of spine head area from eGFP expressing neurons in control (white), gabazine (black), or muscimol (pattern). Ctrl $n = 1301$ spines; Gbz $n = 1082$ spines; Musc $n = 1014$ spines, 3 cultures; Ctrl vs. Gbz $p = 0.005$, Ctrl vs. Musc $p = 0.978$. **c** Median values ±25–75% IQR of spine head area from shMock or shWNK1 overexpressing neurons in control (white and black stripes) or gabazine (black and white stripes) conditions, respectively. Note the increase in spine head area upon application of gabazine for shMock ($p < 0.001$) transfected cells and the decrease ($p = 0.009$) in spine head volume for shWNK1 overexpressing neurons. ShMock $n = 260$ spines; shMock + Gbz $n = 232$ spines; shWNK1 $n = 95$ spines, shWNK1 + Gbz $n = 166$ spines, 2 cultures. **d** Same as in **a** for neurons expressing eGFP and KCC2-Flag or KCC2-Flag mutated on T 906/1007 to A (T906/T1007A) or glutamate (T906/T1007E). Scale bars, 10 µm. **e** Overexpression of T906/T1007E mutant (gray) but not T906/T1007A mutant (fine stripe) increased spine head volume compared with WT (white). **f** No change in spine head volume could be observed upon gabazine application in cells expressing KCC2-Flag T906/T1007A (wide stripe). **e, f** T906/T1007 $n = 1301$ spines, T906/T1007A $n = 1224$ spines, T906/T1007E $n = 983$ spines, T906/T1007A + Gbz $n = 1093$ spines, 3 cultures; T906/T1007 vs. T906/T1007E $p = 0.014$, T906/T1007 vs. T906/T1007A $p = 0.099$, T906/T1007A vs. T906/T1007A + Gbz $p = 0.06$. **b, c, e, f** Spine head area in µm$^2$. The KS test was used for all data comparison

not significantly affect the internalization of KCC2. Therefore, gabazine increases the membrane turnover of KCC2, likely due to increased diffusion and cluster dispersal.

Reduced expression of membrane-inserted KCC2 is predicted to reduce Cl$^-$ export and impact GABA$_A$R-mediated transmission. In order to test this, we compared the reversal potential of GlyR currents ($E_{Gly}$) upon application of gabazine in neurons transfected with a recombinant glycine receptor α1 subunit[32]. KCC2 dispersion upon gabazine was accompanied by a significant depolarization of $E_{Gly}$, reflecting an increase in [Cl$^-$]$_i$ (Fig. 8a–c). This gabazine-induced increase in [Cl$^-$]$_i$ is reminiscent of the reduced FRET ratio of SuperClomeleon observed in neurons exposed for 50 min to the drug (Fig. 4b). Therefore, blocking GABA$_A$R-mediated inhibition results in reduced KCC2-dependent Cl$^-$ export due to reduced membrane expression of the transporter.

In addition to regulating intraneuronal [Cl$^-$]$_i$, KCC2 function also controls dendritic spine head volume in mature hippocampal neurons[3]. Thus, KCC2 suppression or chronic pharmacological blockade of its transport activity increases spine head diameter[3, 4]. We therefore tested whether GABA$_A$R-dependent regulation of KCC2 dynamics and membrane stability may also alter dendritic spines in mature neurons. Up to 30 min muscimol application had no significant effect on spine head area (Fig. 9a, b). Gabazine however caused a 1.4-fold increase in spine head area (Fig. 9a, b), consistent with a reduced expression and/or function of the transporter in spines. This gabazine-mediated increase in spine

head area required WNK1 but not WNK3 activity, as it was blocked by shRNAs targeting WNK1 (Fig. 9c) but not WNK3 (Supplementary Fig. 6). Expression of KCC2 T906/T1007A did not affect dendritic spine morphology (Fig. 9d, e). In contrast, spine head size was increased 1.3-fold upon overexpression of the phospho-mimicking KCC2-T906/T1007E, as compared with WT KCC2 (Fig. 9d, e). These results support the notion that most KCC2 is dephosphorylated at T906/T1007 under basal activity conditions in mature neurons and that phosphorylation of these residues reduces KCC2 membrane expression, thereby leading to spine swelling. Moreover, expression of KCC2-T906/T1007A completely blocked gabazine-induced spine head swelling (Fig. 9f). Collectively, these results demonstrate that transient blockade of GABA$_A$R-mediated transmission alters KCC2 surface expression and ion transport function via effects on WNK1 kinase-dependent KCC2 T906/T1007 phosphorylation, thereby affecting both GABA signaling and dendritic spine morphology.

We next asked whether GABA$_A$R-dependent regulation of KCC2 and NKCC1 may occur under physiological and/or pathological conditions. We first tested whether the gabazine-mediated regulation of KCC2 occurred in cultured hippocampal neurons in the absence of sodium channel and glutamate receptor blockers. In such conditions, acute application of gabazine still reduced by ~1.2-fold the mean fluorescence intensity of clusters and the mean fluorescence intensity per pixel within clusters (Supplementary Fig. 11) as compared with untreated cells. Furthermore, this effect was blocked by closantel (Supplementary

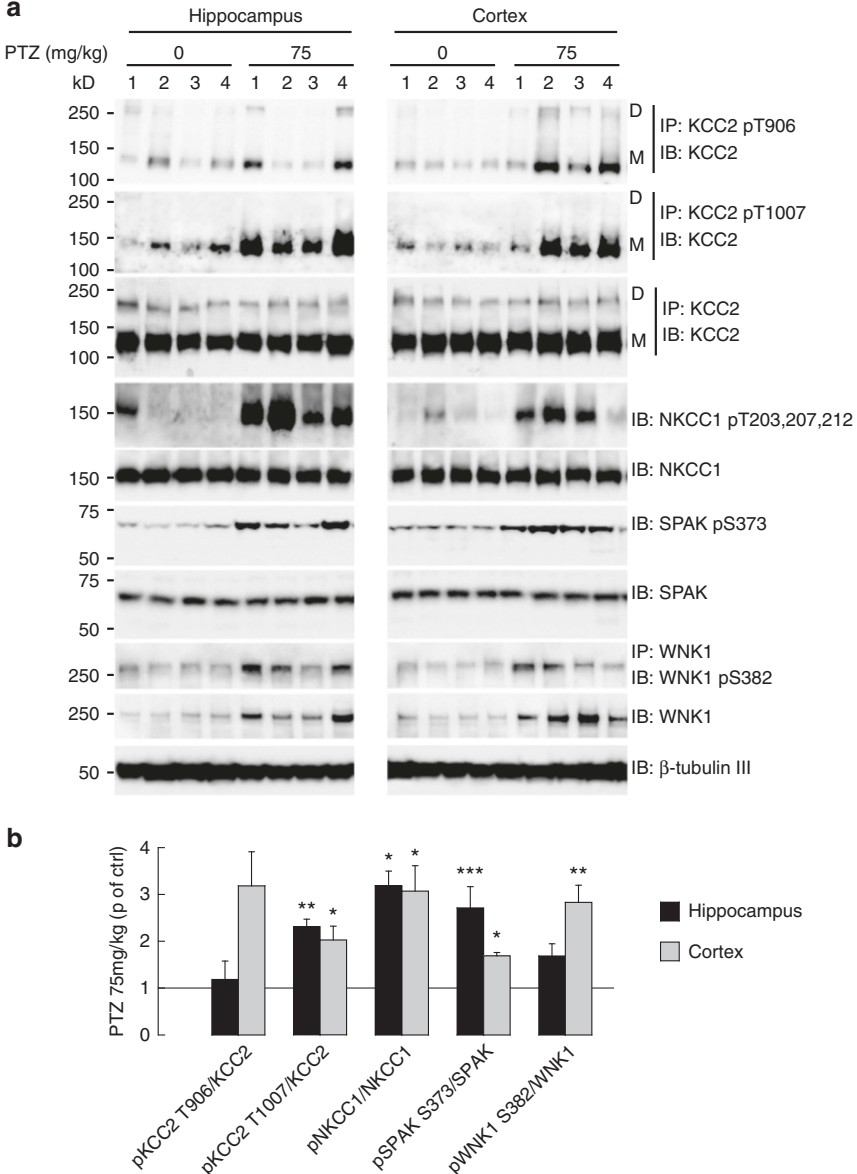

**Fig. 10** GABAAR-dependent regulation of KCC2 in vivo. **a**, **b** Activation of the WNK/SPAK signaling pathway in the adult epileptic brain. Western blot **a** and quantification (**b** mean ± s.e.m., four independent experiments 1–4) showing seizure induction with the GABAAR antagonist pentylenetetrazole (PTZ, 75 mg/kg) led to phosphorylation/activation of WNK1 S382 ($t$-test $p = 0.003$ and $p = 0.071$) and SPAK S373 ($t$-test $p < 0.001$ and $p = 0.012$), respectively, in the cortex (gray) and hippocampus (black) of adult mice. This was accompanied by a net increase in the phosphorylation of NKCC1 T203/T207/T212 (MW test $p = 0.029$) and KCC2T1007 ($t$-test $p = 0.021$ and $p = 0.002$) but not KCC2 T906 (MW test $p = 0.057$ and $p = 0.686$) in the cortex and hippocampus, respectively

Fig. 11), thus involving the WNK/SPAK signaling pathway. These results demonstrate that GABA$_A$R-dependent regulation of KCC2 operates under physiological conditions with intact neuronal and synaptic activity. We then explored the impact of GABA$_A$R activation on KCC2 membrane stability in the intact hippocampal network. Surface biotinylation experiments were performed in acute hippocampal slices prepared from 5–7-week-old C57bl6 mice treated with vehicle or muscimol for 30 min at 35 °C. We found that acute muscimol treatment increased by 1.2 and 1.3-fold the surface expression level of KCC2 monomers and oligomers, respectively (Supplementary Fig. 11). Therefore, we conclude that GABA$_A$R activation stabilizes KCC2 at the neuronal surface in the intact hippocampal network.

Finally, we explored whether the WNK1/SPAK/OSR1 signaling pathway could be activated and regulate KCC2/NKCC1 threonine phosphorylation in vivo. The activity of the WNK1/SPAK/OSR signaling pathway is elevated in the embryonic and neonatal first postnatal week and significantly decreases with brain maturation into adulthood[17]. We therefore asked whether this pathway could be "reactivated" in the mature brain under pathological conditions. Upon subcutaneous injection of the convulsant, GABA$_A$R antagonist pentylenetetrazole (PTZ)[33], we observed massive changes in both WNK1 and SPAK/OSR1 phosphorylation/activation (by 169–283% and 271–169%, respectively), as well as KCC2 T906, KCC2T1007, and NKCC1 T203/T207/T212 phosphorylation (by 118–318%, 232–203% and 319–307%, Fig. 10a, b, Supplementary Fig. 12) in both cortex and hippocampus. These data demonstrate the WNK1/SPAK/OSR1 signaling pathway can be activated in the adult brain upon

impaired $GABA_A R$ activity to simultaneously upregulate NKCC1 and downregulate KCC2.

## Discussion

Whereas KCC2 is known to be rapidly downregulated by enhanced neuronal activity and glutamatergic neurotransmission in mature neurons[10, 12, 14], only one study so far had tested the effect of GABA signaling on KCC2 post-translational regulation. This study showed that increased GABAergic transmission leads to KCC2 downregulation[34]. However, this study was carried out in immature neurons with depolarizing $GABA_A R$-mediated responses and associated activation of VDCC and intracellular $Ca^{2+}$ signaling. In the present study, we asked instead whether KCC2 may be regulated by $GABA_A R$-mediated inhibition in mature neurons. Our results reveal a rapid regulation of KCC2 membrane trafficking and transporter function by $GABA_A R$-dependent inhibition. KCC2 lateral diffusion, membrane clustering, and stability are regulated by $GABA_A R$- but not $GABA_B R$-mediated signaling. $GABA_A R$ activation leads to KCC2 confinement, while $GABA_A R$ blockade increases KCC2 membrane dynamics, reducing its membrane aggregation, stability and activity. Our data uncover a novel homeostatic mechanism that may serve for "auto-tuning" of GABAergic signaling via rapid regulation of KCC2-mediated $Cl^-$ export. Our work also demonstrates that $Cl^-$ ions may act as genuine intracellular second messengers in neurons to modulate KCC2 phosphorylation through the WNK signaling pathway.

Different subpopulations of KCC2 exist in the plasma membrane, including freely diffusing transporters that are uniformly distributed at the cell surface and slower, confined transporters within membrane clusters. Watanabe et al.[35] provided the first evidence suggesting clustering may be directly involved in KCC2 function. Tyrosine mutation in this study was shown to reduce KCC2 aggregation together with a significant reduction in ion transport without significant change in membrane stability. Our present results instead support the notion that rapid changes in KCC2 membrane dynamics and clustering directly impact KCC2 membrane stability and neuronal $Cl^-$ homeostasis. We have shown that glutamate receptor activation[10] or blockade of $GABA_A Rs$ (this study) both lead to an increase in KCC2 lateral diffusion. In both cases, increased KCC2 diffusion was correlated with a change in KCC2 phosphorylation (S940 dephosphorylation upon increased excitation; T906/T1007 phosphorylation upon $GABA_A R$ blockade). Phosphorylation-dependent alteration of membrane diffusion therefore represents one of the first mechanisms regulating KCC2 membrane expression and function. We propose as a working model that T906/T1007 phosphorylation as well as S940[10] dephosphorylation may induce conformational changes[36] leading to altered interaction with scaffolding molecules or oligomerization of the transporter that in turn may promote cluster dispersion. Freely diffusing transporters may then become available for internalization.

Blockade of $GABA_A R$-mediated transmission and increased neuronal activity both lead to an increase in KCC2 lateral diffusion. While neuronal excitation modulates KCC2 diffusion and membrane stability via $Ca^{2+}$-dependent dephosphorylation of S940[10, 12], we show that neuronal inhibition regulates KCC2 membrane dynamics independently of $Ca^{2+}$ and S940. Instead, $Cl^-$-dependent changes in WNK1 kinase activity link $GABA_A R$ signaling to T906/T1007 phosphorylation (Supplementary Fig. 13).

Although WNK1 is activated by low $[Cl^-]_i$, it is however more active in immature neurons in which $[Cl^-]_i$ is higher than in mature neurons[17, 26, 37]. This apparent paradox may reflect the expression pattern of WNK3, which is inversely correlated to that

of KCC2 during neuronal development[38]. WNK3 shows weak sensitivity to chloride and is able to activate WNK1[25, 39]. Strong expression of WNK3 in immature neurons and subsequent WNK1 activation may then render WNK1 insensitive to $[Cl]_i$ in immature neurons. Reduced expression of WNK3 in mature neurons would then enable regulation of WNK1 activity by changes in $[Cl]_i$.

Activation of GABAergic synapses leads to changes in $[Cl^-]_i$[40]. Changes in the activity of GABAergic synapses could hence rapidly alter KCC2 membrane expression through WNK1-dependent KCC2 phosphorylation. Changes in $GABA_A R$ activation and a subsequent increase in KCC2 confinement would maintain KCC2-mediated $Cl^-$ extrusion and help counteract $Cl^-$ influx through $GABA_A Rs$.

We propose that gabazine lowers $[Cl]_i$ by decreasing $Cl^-$ influx through $GABA_A R$, and this reduces KCC2 activity via WNK1-dependent regulation of its membrane stability. However, we showed an increase, not a decrease in $[Cl]_i$ (Figs. 4b, 8) after neuronal exposure to gabazine. This may be due to the fact that changes in $[Cl]_i$ upon gabazine application occur on different time scales. Although SuperClomeleon imaging could not detect an immediate drop in $[Cl]_i$ upon gabazine application, we observed a delayed increase in $[Cl]_i$ that is consistent with our electrophysiological recordings. We propose that gabazine application leads to a transient drop in $[Cl^-]_i$ that in turn activates the WNK/SPAK/OSR signaling pathway leading to KCC2 phosphorylation on T906 and T1007. This effect is likely to persist well beyond the initial drop in $[Cl]_i$ as it will be reversed through the recruitment of PP1 phosphatase[41, 42]. Persistent KCC2 inactivation by threonine phosphorylation would then be expected to translate into a rise in $[Cl^-]_i$, consistent with our electrophysiological and chloride imaging data.

$GABA_A R$-dependent regulation of KCC2 may then not only allow neurons to rapidly react to changes in $[Cl^-]_i$ but also permit the neuron to conserve energy. Indeed, for every $Cl^-$ ion extruded by KCC2, the transporter uses the energy of the electrochemical gradient of one $K^+$ ion. The $Na^+/K^+$ ATPase that generates the $K^+$ gradient required for KCC2 function is a major energy consumer in the brain[43–45]. Even though the highest energetic cost of the $Na^+/K^+$-ATPase will be used to restore the resting potential of the cell after neuronal firing[44, 46], maintaining low $[Cl^-]_i$ is associated with high metabolic cost[31]. Under physiological conditions, rapid redistribution of KCC2 in the membrane might enable neurons to conserve energy by keeping surface KCC2 molecules at the minimum required density to keep $E_{GABA}$ hyperpolarized. Diffusion-trap of KCC2 is hence a very rapid mechanism to adjust the number of KCC2 molecules in the membrane to ensure functional $Cl^-$ homeostasis in neurons.

Finally, our biochemical data indicate that modulation of $GABA_A R$ activity translates not only into a change in KCC2 but also NKCC1 phosphorylation. Given the opposite effect of phosphorylation on KCC2 and NKCC1 activity (activating for NKCC1, inhibitory for KCC2), regulation of the WNK-SPAK pathway is a very efficient mechanism to adjust neuronal $Cl^-$ homeostasis. Impairment of this pathway has been linked to dysregulated $Cl^-$ homeostasis in schizophrenia, autism, and epilepsy[47–50]. We further demonstrate here that this signaling pathway is rapidly and massively activated in an acute epilepsy model. This pathway could therefore be an attractive target to restore $Cl^-$ homeostasis for therapeutic benefit.

## Methods

**Neuronal culture**. All animal procedures were carried out according to the European Community Council directive of 24 november 1986 (86/609/EEC), the guidelines of the French Ministry of Agriculture and the Direction Départementale de la Protection des Populations de Paris (Institut du Fer à Moulin, Animalerie des

Rongeurs, license C 72-05-22). All efforts were made to minimize animal suffering and to reduce the number of animals used.

Primary cultures of hippocampal neurons were prepared as previously described[10] with some modifications of the protocol. Briefly, hippocampi were dissected from embryonic day 18 or 19 Sprague-Dawley rats of either sex. Tissue was then trypsinized (0.25% v/v), and mechanically dissociated in 1× HBSS (Invitrogen, Cergy Pontoise, France) containing 10 mM HEPES (Invitrogen). Neurons were plated at a density of $120 \times 10^3$ cells/ml onto 18-mm diameter glass coverslips (Assistent, Winigor, Germany) pre-coated with 50 µg/ml poly-D,L-ornithine (Sigma-Aldrich, Lyon, France) in plating medium composed of Minimum Essential Medium (MEM, Sigma) supplemented with horse serum (10% v/v, Invitrogen), L-glutamine (2 mM) and Na$^+$ pyruvate (1 mM) (Invitrogen). After attachment for 3–4 h, cells were incubated in culture medium that consists of Neurobasal medium supplemented with B27 (1X), L-glutamine (2 mM), and antibiotics (penicillin 200 units/ml, streptomycin, 200 µg/ml) (Invitrogen) for up to 4 weeks at 37 °C in a 5% $CO_2$ humidified incubator. Each week, one fifth of the culture medium volume was renewed.

**Neuro2a culture**. Neuro2a cells were obtained from DSMZ (ACC-148). Their origin was confirmed by COI DNA barcoding and they have been tested for mycoplasma contamination by DAPI, microbiological culture, RNA hybridization and PCR assays. Neuro2a cells were grown in DMEM GlutaMAX (Invitrogen) supplemented with 1 g/L glucose and 10% fetal bovine serum.

**DNA constructs**. The KCC2-Flag construct was generated by insertion of three Flag sequences in the second predicted extracellular loop of KCC2[10]. This recombinant Flag-tagged KCC2 transporter was shown to retain normal traffic and function in transfected hippocampal neurons[10]. KCC2-Flag constructs with Threonine residues 906 and 1007 mutated to glutamate (T906/1007E) or alanine (T906/1007 A) were generated by Genscript (Piscataway, USA). Threonine nucleotide sequence was changed to GCA for alanine substitution and to GAG for glutamate substitution. cDNA from E18.5 rat brain was used as a template for the amplification of WNK3 coding transcript. Primers were chosen to amplify three overlapping fragments using Phusion DNA polymerase (Thermofisher Scientific). To obtain N-terminal HA-tagged WNK3, the purified PCR fragments were directionally assembled to a pCAGGS-HA linearized plasmid using In-Fusion HD cloning kit (Clontech) following manufacturer's guidelines. ShRNAs against rat WNK3 were designed and cloned into pGeneclipU1(GFP) (Promega) following manufacturer's protocol. The following target sequences were used: shRNA #1: 5′-GAACCTTAAAGACGTACTTAA-3′ and shRNA #2: 5′-GAACGCCTTCGAG-CAACTAAA-3′. All the constructs were sequence-verified by Beckman Coulter Genomics. The following constructs were also used: KCC2-Flag-S940D[10], eGFP (Clontech), glycine receptor α1 subunit[32] (kindly provided by A. Triller, ENS, Paris), gephyrin-mRFP[51] (kindly provided by A. Triller, ENS, Paris), homer1c-GFP[52] (kindly provided by D. Choquet, IIN, Bordeaux, France), WNK1 with "kinase-dead, dominant-negative domain" (WNK1-KD, D368A), "constitutively active" WNK1 (WNK1-CA, S382E), shRNA against rat WNK1 mRNA (shWNK1), a scrambled shRNA sequence "shMock"[17] (kindly provided by I. Medina, INMED, Marseille) and SuperClomeleon[22] (kindly provided by G.J. Augustine, NTU, Singapore). All constructs were sequenced by Beckman Coulter Genomics (Hope End, Takeley, U.K).

**Transfection and transduction**. Neuronal transfections were carried out at DIV 13–14 using Lipofectamine 2000 (Invitrogen) or Transfectin (BioRad, Hercules, USA), according to the manufacturers' instructions (DNA:transfectin ratio 1 µg:3 µl), with 1–1.5 µg of plasmid DNA per 20 mm well. The following ratios of plasmid DNA were used in co-transfection experiments: 0.5:0.4:0.3 µg for KCC2-Flag or KCC2-Flag-T906/T1007A or KCC2-Flag-T906/T1007E or KCC2-Flag-S940D together with gephyrin-mRFP and homer1c-GFP; 0.9:0.1 µg for KCC2-Flag or KCC2-Flag-T906/T1007A or KCC2-Flag-T906/T1007E with eGFP or glycine receptor α1 subunit with eGFP; 0.5:0.5 µg for KCC2-Flag or KCC2-Flag-T906/T1007A or KCC2-Flag-T906/T1007E with shWNK1 or shMock or WNK1-KD or WNK1-CA or KCC2-Flag with SuperClomeleon or KCC2-Flag with shNT; 0.5:0.5:0.5 µg for KCC2-Flag with shWNK3 #1 and shWNK3 #2. Experiments were performed 7–10 days post-transfection.

Primary hippocampal cultures were infected at DIV 14 with an AAV1.hSyn. eNpHR3.0-eYFP.WPRE.hGH (Title = 1.6 e$^{12}$, Upenn Vector Core) at a multiplicity of infection (MOI) of 10. The pAAV-hSyn-eNpHR 3.0-EYFP was a gift from Karl Deisseroth (Addgene plasmid #26972). Neurons were used 7 days after transduction.

Neuro2a cells were transfected using Transfectin (Biorad) according to the manufacturers' instructions (DNA:transfectin ratio 1 µg:3 µl), with 1–1.5 µg of plasmid DNA per 60 mm well. Cells were transfected with 1 µg of SuperClomeleon; 0.5:0.5 µg of KCC2-Flag with shNT or shWNK3 #1 or shWNK3 #2; 0.5:0.5:0.5 µg of KCC2-Flag with shWNK3 #1 and shWNK3 #2. Protein expression was allowed in growth medium for 48–72 h after transfection before use.

**Pharmacology**. The following drugs were used: TTX (1 µM; Latoxan, Valence, France), R,S-MCPG (500 µM; Abcam, Cambridge, UK), Kynurenic acid (1 mM;

Abcam), NBQX (10 µM; Tocris), R,S-APV (100 µM; Tocris), VU0240551 or VU0463271 (10 µM; Sigma), CGP52432 (20 µM; Tocris Bioscience, Lille, France), baclofen (20 µM; Tocris), gabazine (10 µM; Abcam), muscimol (10 µM; Abcam), L655,708 (50 µM; Tocris Bioscience), closantel (10 µM; Sigma), picrotoxin (100 µM; Tocris), Cadmium chloride (100 µM; Sigma). R,S-MCPG and baclofen were prepared in equimolar concentrations of NaOH; TTX in 2% citric acid (v/v); VU0240551, closantel and picrotoxin in DMSO (Sigma). Equimolar DMSO concentrations were used for control experiments in these conditions. For SPT experiments, neurons were transferred to a recording chamber, pre-incubated in presence of drugs at 31 °C for 10 min in imaging medium (see below for composition) and used within 45 min in presence of the appropriate drug for imaging. For calcium imaging in presence of TTX + KYN + MCPG + Cd$^{2+}$, cells were pre-incubated 5–10 min in presence of these drugs in imaging medium during the Fluo4-AM hydrolysis, and gabazine was applied after a stable fluorescence baseline was obtained. For calcium imaging in absence of TTX + KYN + MCPG + Cd$^{2+}$, cells were pre-incubated 5–10 min in imaging medium during the Fluo4-AM hydrolysis, and NMDA was applied after a stable fluorescence baseline was obtained. For biochemistry and immunofluorescence experiments, drugs were added directly to the culture medium for 30 min in a $CO_2$ incubator set at 37 °C. The imaging medium consisted of phenol red-free minimal essential medium supplemented with glucose (33 mM; Sigma) and HEPES (20 mM), glutamine (2 mM), Na$^+$-pyruvate (1 mM), and B27 (1×) from Invitrogen. The 138 mM [Cl$^-$] extracellular solution was composed of 2 mM CaCl$_2$, 2 mM KCl, 3 mM MgCl$_2$, 10 mM HEPES, 20 mM glucose, 126 mM NaCl, 15 mM Na methanesulfonate; the 0 mM [Cl$^-$] extracellular solution was made of 1 mM CaSO$_4$, 2 mM K methanesulfonate, 2 mM MgSO$_4$, 10 mM HEPES, 20 mM glucose, 144 mM Na methanesulfonate.

**Live cell staining for single-particle imaging**. Neurons were stained as described previously[53]. Briefly, cells were incubated for 3–5 min at 37 °C with primary antibodies against Flag (mouse, 1:700, Sigma, cat #F3165), washed, and incubated for 3–5 min at 37 °C with biotinylated Fab secondary antibodies (goat anti-mouse: 1:700; Jackson Immuno research, cat #115-067-003, West Grove, USA) in imaging medium. After washes, cells were incubated for 1 min with streptavidin-coated quantum dots (QDs) emitting at 605 nm (1 nM; Invitrogen) in borate buffer (50 mM) supplemented with sucrose (200 mM) or in PBS (1 M; Invitrogen) supplemented with 10% Casein (v/v) (Sigma).

**Single-particle tracking and analysis**. Cells were imaged as previously described[10] using an Olympus IX71 inverted microscope equipped with a 60X objective (NA 1.42; Olympus) and a Lambda DG-4 monochromator (Sutter Instrument) or a 120 W Mercury lamp (X-Cite 120Q, Lumen Dynamics). Individual images of gephyrin-mRFP and homer1c-GFP, and QD real time recordings (integration time of 30 ms over 1200 consecutive frames) were acquired with an ImagEM EMCCD camera and MetaView software (Meta Imaging 7.7). Cells were imaged within 45 min following appropriate drugs pre-incubation.

QD tracking and trajectory reconstruction were performed with homemade software (Matlab; The Mathworks, Natick, MA) as described in refs [10, 53]. One to two sub-regions of dendrites were quantified per cell. In cases of QD crossing, the trajectories were discarded from analysis. Trajectories were considered synaptic when overlapping with the synaptic mask of gephyrin-mRFP or homer1c-GFP clusters, or extrasynaptic for spots two pixels (380 nm) away[54]. Values of the mean square displacement (MSD) plot vs. time were calculated for each trajectory by applying the relation:

$$\mathrm{MSD}(n\tau) = \frac{1}{N-n} \sum_{i=1}^{N-n} \left[ \left( x_{(i+n)} - x_i \right)^2 + \left( y_{(i+n)} - y_i \right)^2 \right],$$

where $\tau$ is the acquisition time, $N$ is the total number of frames, $n$ and $i$ are positive integers with $n$ determining the time increment. Diffusion coefficients (D) were calculated by fitting the first four points without origin of the MSD vs. time curves with the equation: $\mathrm{MSD}(n\tau) = 4Dn\tau + b$, where $b$ is a constant reflecting the spot localization accuracy. Depending on the type of lamp used for imaging, the QD pointing accuracy is ~20–30 nm, a value well below the measured explored areas (at least 1 log difference). Synaptic dwell time was defined as the duration of detection of QDs at synapses on a recording divided by the number of exits as detailed previously[55, 56]. The explored area of each trajectory was defined as the MSD value of the trajectory at two different time intervals of at 0.42 and 0.45 s[57]. The number of QDs vary from one cell to another and from one condition to another in a given experiment. To avoid giving too much weight to a condition with larger numbers and in order to compare the conditions between them within the same culture, the number of QDs in each condition was adjusted to the number of QDs of the condition with the smallest number. For conditions with large numbers, values were sorted randomly using the ALEA function of Excel 2013 before data extraction.

**Chloride imaging**. Calibration of SuperClomeleon was performed in Neuro2a cells. Extra and intracellular chloride were equilibrated using the K$^+$/H$^+$ ionophore nigericin and the Cl/OH antiporter tributyltin as described[22]. Cells were perfused

with a solution containing nigericin (10 μM, Sigma), tributyltin chloride (10 μM, Sigma) and (in mM) EGTA 2, κ-gluconate 2, MgCl₂/Mg sulfonate 2, HEPES 10, glucose 20 and NaCl/Na gluconate ranging from 0 to 138 (pH 7.4). Neuro2a cells and neurons were imaged at 35 °C in a temperature-controlled open chamber (BadController V, Luigs and Neumann, Ratingen, Germany). Two-photon imaging was performed using an upright LeicaTCS MP5 microscope equip with resonant scanner (8 kHz), a Leica ×25/0.95 HCX IRAPO immersion objective and a tunable Ti:sapphire laser (Coherent Chameleon Vision II) with dispersion correction set to 820 nm for CFP excitation. The emission path consisted of an initial 700 nm low-pass filter to remove excess excitation light (E700 SP, Chroma Technologies), 506 nm dichroic mirror for orthogonal separation of emitted signal, 483/32 CFP emission filter, 535/30 YFP emission filter (FF506-Di01-25 36; FF01-479/40; FF01-542/50; Brightline Filters; Semrock) and two-channel Leica HyD detector for simultaneous acquisition. For Neuro2a imaging, due to the high-expression and low-dark noise of the HyD photodetectors, detector gain was typically set at 10% with a laser power at 5%. For neurons imaging, due to the expression variability and lower expression of the sensor compared with Neuro2a transfection, detector gain of the HyD photodetectors was typically set at 50–100% with laser power at 7%. Z-stack images (16-bit; 512 × 512) were typically acquired every 30 s. The z-step size was 1 μm and total stack size was typically 20–30 sections depending on the neurons. Images were processed in ImageJ by using maximum z-projections followed by the correction of minor coverslip drifts using StackReg macro[58]. Regions of interest (ROIs) were selected for measurement if they could only be measured over the whole experiment. Raw CFP and YFP intensity measurements for the entire recording were imported into Microsoft Excel. Background fluorescence (measured from a non-fluorescent area) was subtracted and a fluorescence ratio was calculated for each time point in each ROI series and was normalized to the average baseline ratio for each respective ROI (average of the first four frames before treatment). Statistical analysis was performed in SigmaPlot 12.5. Wilcoxon rank-sum test (or paired t-test when normality test passed) was used to compare the mean responses before and upon pharmacological treatment.

**Calcium imaging**. Neurons at DIV21-25 were loaded with 0.5 mM Fluo-4AM (Invitrogen) for 5 min at 37 °C in imaging medium. After washing excess dye, cells were further incubated for 5–10 min to allow hydrolysis of the AM ester. Cells were imaged at 37 °C in an open chamber mounted on an inverted spinning-disc microscope (Leica DMI4000, Yokogawa CS20 spinning Nipkow disk, ×40/0.6 N.A. objective). All washes, incubation steps, and cell imaging were performed in imaging medium. Fluo4-AM was illuminated using 491 nm light from a diode. Emitted light was collected using a 525-39 (±25) nm emission filter. Time lapse (0.33 Hz for 600 s) of confocal stacks (of ~35 images acquired with an interval of 0.3 μm) were acquired with a cooled EM-CCD camera (512 × 512, 16 μm pixel size) using Metamorph. The analysis was performed on a section of the stack where the soma was in focus at different time points. Fluorescence intensities collected in the soma before (F0) and following (F) bath addition of the drugs, were background-subtracted before being displayed as F/F0 values. The data were analyzed using Metamorph. Normalization of fluorescence intensity was performed for each cell by dividing the mean fluorescence intensity by the average of fluorescence intensities of the four time points before drug application. Statistics (paired t-test) were run on the last time point before drug application (120 s) compared with the latest time point after drug application (600 s).

**Immunocytochemistry**. KCC2-Flag membrane clustering was assessed with live cell staining. Pre-treated neurons expressing KCC2-Flag were washed in imaging medium and incubated for 20 min at 4 °C with mouse primary antibody against Flag (1:400; Sigma, cat #F3165) in imaging medium in the presence of the appropriate drugs. After washes with imaging medium, cells were fixed for 15 min at room temperature (RT) in paraformaldehyde (PFA; 4% w/v; Sigma) and sucrose (20% w/v; Sigma) solution in 1× PBS. The cells were then washed in PBS and incubated for 30 min at RT in goat serum (GS; 20% v/v; Invitrogen) in PBS to block non-specific staining. Neurons were then incubated for 45 min at RT with Cy5-conjugated goat anti-mouse antibodies (1.9 g/ml; Jackson Immuno Research, cat #115-175-205) in PBS–GS blocking solution, washed, and mounted on slides. To assess spine morphology pre-treated KCC2-Flag and eGFP co-transfected cells were fixed for 15 min at room temperature (RT) and washed in 1× PBS. Coverslips were mounted on slides with mowiol 844 (48 mg/ml; Sigma).

**Fluorescence image acquisition and analysis**. Image acquisition was performed using a ×100 objective (NA 1.40) on a Leica (Nussloch, Germany) DM6000 upright epifluorescence microscope with a 12-bit cooled CCD camera (Micromax, Roper Scientific) run by MetaMorph software (Roper Scientific, Evry, France). Quantification was performed using MetaMorph software (Roper Scientific). Image folders were randomized before analysis. For morphological spine analysis exposure time was adjusted for each eGFP image to obtain best fluorescence to noise ratio and to avoid pixel saturation. For each neuron a well-focused dendrite was chosen, spine heads were manually delimited and their area were quantified. To assess KCC2-Flag clusters, exposure time was fixed at a non-saturating level and kept unchanged between cells and conditions. For cluster analysis, images were first flatten background filtered (kernel size, 3 × 3 × 2) to enhance cluster outlines, and a user

defined intensity threshold was applied to select clusters and avoid their coalescence. Clusters were outlined and the corresponding regions were transferred onto raw images to determine the mean KCC2–Flag cluster number, area and fluorescence intensity. The dendritic surface area of the region of interest was measured to determine the number of clusters per 10 μm². For each culture, we analyzed ~10 cells per experimental condition and ~100 clusters or ~15 spines per cell.

**Total RNA extraction and cDNA synthesis**. Total RNA extraction was performed on DIV 21 hippocampal cultures using RNeasy Mini kit (Qiagen, Venlo, The Netherlands) and on Sprague-Dawley rat (13 weeks) hippocampus tissue using RNAsolv (Omega Bio-tek, Norcross, GA) according to manufacturer's instructions. Genomic DNA was removed by digestion with Amplification Grade DNase I (Sigma-Aldrich). First-strand cDNA was synthesized by reverse transcription of 1 μg of total RNA using Superscript-II and random primers (Invitrogen) according to standard protocols. Reverse transcriptase was absent in some samples as negative control.

**Quantitative real-time PCR**. Relative expression levels of WNK1-4 mRNA were determined by real time RT-PCR using Absolute SYBR Green Mix (ABgene, Epsom, UK) and the following set of primers: WNK1-for (5′-AAGGTCTGGA-CACCGAAACC-3′), WNK1-rev (5′-TTCCCTTTTACTGTGGATTCCC-3′), WNK2-for (5′-CATGACATGGAGGCCTCTGG-3′), WNK2-rev (5′-CGGGCT TTTCACTCTCAGGA-3′), WNK3-for (5′-CATCACAGGACCCACTGGAT-3′), WNK3-rev (5′-AGCCATTTCCAACATACACATC-3′), WNK4-for (5′-GCTGCA AACTCACAACAGCA-3′), and WNK4-rev (5′-CTCAGGAATCCGTCTCG CTC-3′). Data were analyzed with the 2−DCt method, and values normalized to WNK1 expression level.

**Surface biotinylation in cultures**. Pre-treated neuronal cultures were washed with ice-cold PBS three times, and then incubated in freshly prepared PBS containing 0.5–1 mg/ml EZ-Link Sulfo-NHS-SS-Biotin (Pierce, Rockford, IL, USA) at 4 °C for 30 min with gentle agitation. Biotinylation was stopped by addition of Tris-HCl (50 mM; pH 7.4) and cells lysed in in modified RIPA buffer (50 mM Tris-HCl (pH 7.4), 150 mM NaCl, 1% Nonidet P-40, 0.5% DOC, 0.1% SDS, 50 mM NaF, 1 mM Na₃VO₄ and protease and phosphatase inhibitors, Roche). After thoroughly homogenizing, the samples were centrifuged and the supernatant collected. A small fraction of the lysates was kept for input KCC2 quantification. Lysates were mixed with 50% slurry of Neutravidin beads (Thermo Scientific) and rotated overnight at 4 °C. The beads were collected by centrifugation and washed three times in modified RIPA buffer and one time in modified RIPA buffer without detergents. After the last wash all solution was carefully removed from the beads, and the biotin-bound and input fractions denatured in 6× SDS sample buffer containing DTT at 37 °C for 1 h.

Samples were subjected to electrophoresis on polyacrylamide gels and transferred to nitrocellulose membranes. The membranes were incubated for 30 min with Tris-buffered saline, with 1% triton (TTBS; Invitrogen) containing 5% (w/v) skim milk. The membranes were then immunoblotted in 5% (w/v) skim milk in TTBS with rabbit anti-KCC2 (Millipore, cat #07-432) and mouse anti-TUJ1 (R&D Systems, cat #MAB 1195) or anti-actin (SantaCruz, cat #sc-32251) antibodies overnight at 4 °C. The blots were then washed four times with TTBS and incubated for 1 h at room temperature with secondary fluorescent antibodies (DyLight700 cat #610-730-002 or 800 cat #611-145-002, Rockland) diluted 5000-fold in 5% (w/v) skim milk in TTBS. After repeating the washing steps, fluorescence was detected using Odyssey infrared imaging system (LI-COR Bioscience). The relative intensities of immunoblot bands were determined by densitometry with ImageJ software. Total KCC2 protein expression was determined as the sum of monomeric and oligomeric bands normalized to TUJ1 or actin. Surface expression of KCC2 was determined as the ratio of monomeric + oligomeric biotinylated KCC2 fraction vs. total KCC2 fraction.

**Surface biotinylation in hippocampal slices**. Biotinylation studies were performed as previously described[59]. Horizontal slices (500 μm) were made from 5–7-week-old wild-type animals (C57Bl/6j, Janvier) using a Leica vibratome in NMDG based cutting solution (in mM: NMDG 93, HCl 93, KCl 2.5, NaH₂PO₄ 1.2, NaHCO₃ 30, HEPES 20, glucose 25, ascorbic acid 5, sodium pyruvate 3, MgCl₂ 10, CaCl₂ 0.5, saturated with 95% O₂/5% CO₂, pH 7.4, 300 mOsm). Slices were transferred into an interface chamber at 37 °C for 10 min containing ACSF (in mM: CaCl₂ 1.6, glucose 11, KCl 2.5, MgCl₂ 1.2, NaCHO₃ 26.2, NaH₂PO₄ 1, NaCl 124, saturated with 95% O₂/5% CO₂, pH 7.4, 298 mOsm), followed by a 1 h recovery period at room temperature. The slices were then placed onto a pre-heated recording chamber in either pre-warmed bubbled ACSF or ACSF containing muscimol (10 μM) for 30 min in ACSF at 35 °C. The slices were transferred into bubbled ice-cold ACSF containing 1 mg/ml EZ-Link Sulfo-NHSSS-Biotin (21326, Thermo Scientific) with gentle rotation for 45 min at 4 °C. Excess biotin was quenched using 1 M glycine in ice-cold ACSF for 10 min, and then the slices were rinsed once in ice-cold ACSF and snap frozen on dry ice. The hippocampus was micro-dissected and immediately lysed and homogenized in modified RIPA buffer (50 mM Tris-HCl (pH 7.4), Triton X-100 1%, 150 mM NaCl, 1 mM EDTA, DOC 0.5%, NP40 1%, SDS 0.1%, 50 mM NaF, complete protease inhibitor cocktail

(Roche)). The samples were centrifuged at 15,000 r.p.m. for 15 min, the supernatant was collected and protein content was determined using a Pierce BCA protein quantification kit (23227, Thermo Scientific). 50 μg of protein was loaded onto 100 μl of 50% slurry of Pierce NeutrAvidin UltraLink Resin (53150, Thermo Scientific), made up to a total volume of 400 μl in modified RIPA buffer and rotated for 2 h at 4 °C. The beads were recuperated by centrifugation and thoroughly washed four times in modified RIPA buffer, and after the last wash the beads were incubated in ×6 SDS sample buffer containing 10% β-mercaptoethanol at 37 °C for 1 h. The protein samples were run on pre-cast Bis-Tris gels (NuPage 4–12% gradient gels, NP0322, Invitrogen) and immunoblotting was performed. Analysis was performed using ImageJ by normalizing the amount of surface KCC2 to the amount of actin in the non-biotinylated fraction.

**Immunoprecipitation and immunoblotting.** Pretreated hippocampal neurons in culture (DIV 23) were lysed in lysis buffer containing 50 mM Tris/HCl, pH 7.5, 1 mM EGTA, 1 mM EDTA, 50 mM sodium fluoride, 5 mM sodium pyrophosphate, 1 mM sodium orthovanadate, 1% (w/v) Nonidet P-40, 0.27 M sucrose, 0.1% (v/v) 2-mercaptoethanol, and protease inhibitors (Roche). CCCs phosphorylated at the KCC2 T906 and T1007 equivalent residue were immunoprecipitated (centrifuged at 16,000×g at 4 °C for 20 min) using phosphorylation site-specific antibodies as described[27]. The phosphorylation site-specific antibodies were coupled with protein-G–Sepharose at a ratio of 1 mg of antibody per 1 ml of beads in the presence of 20 μg/ml of lysate to which the corresponding non-phosphorylated peptide had been added. Two milligrams of clarified cell lysate was incubated with 15 μg of antibody conjugated to 15 μl of protein-G–Sepharose for 2 h at 4 °C with gentle agitation. Beads were washed three times with 1 ml of lysis buffer containing 0.15 M NaCl and twice with 1 ml of wash buffer (50 mM Tris/HCl, pH 7.5 and 0.1 mM EGTA). Bound proteins were eluted with 1× LDS sample buffer (Invitrogen) containing 1% (v/v) 2-mercaptoethanol.

Cell or tissue lysates (15 μg) in SDS sample buffer were subjected to electrophoresis on polyacrylamide gels and transferred to nitrocellulose membranes. The membranes were incubated for 30 min with TTBS containing 5% (w/v) skim milk. The membranes were then immunoblotted in 5% (w/v) skim milk in TTBS with the indicated primary antibodies overnight at 4 °C. Antibodies prepared in sheep[17] were used at a concentration of 1–2 μg/ml. The incubation with phosphorylation site-specific sheep antibodies was performed with the addition of 10 μg/ml of the non-phosphorylated peptide antigen used to raise the antibody. The blots were then washed six times with TTBS and incubated for 1 h at room temperature with secondary HRP-conjugated antibodies diluted 5000-fold in 5% (w/v) skim milk in TTBS. After repeating the washing steps, the signal was detected with the enhanced chemiluminescence reagent. Immunoblots were developed using a film automatic processor (SRX-101; Konica Minolta Medical) and films were scanned with a 600-dpi resolution on a scanner (PowerLook 1000; UMAX). The relative intensities of immunoblot bands were determined by densitometry with ImageJ software.

Antibodies used for western blots were raised in sheep and affinity-purified on the appropriate antigen by the Division of Signal Transduction Therapy Unit (DSTT) at the University of Dundee; other antibodies were purchased. KCC1 total antibody (S699C, first bleed; raised against residues 1–118 of human KCC1); KCC2 total antibody (S700C, first bleed; raised against residues 1–119 of human KCC2A); KCC3 total antibody (S701C, first bleed; raised against residues 1–175 of human KCC3); KCC4 total antibody (S801C, first bleed; raised against residues 1–117 of human KCC4); KCC2a phosphoT906 (S959C, first bleed; raised against residues 975–989 of human KCC3a phosphorylated at T991, SAYTYER(T)LMMEQRSRR); KCC2a phosphoT1007 (S961C, first bleed; raised against residues 1032–1046 or 1041–1055 of human KCC3a phosphorylated at T1048). NKCC1 total antibody (S022D, second bleed; raised against residues 1–288 of human NKCC1); NKCC1 phospho-T203/T207/T212 antibody (S763B, third bleed; raised against residues 198–217 of human NKCC1 phosphorylated at T203, T207, and T212, HYYYD(T)HTN(T)YYLR(T)FGHNT); WNK1-total antibody (S079B, second bleed; raised against residues 2360–2382 of human WNK1); WNK1phospho-S382 antibody WNK1-phospho-Ser[382] antibody (S099B, second bleed; raised against residues residues 377–387 of human WNK1 phosphorylated at S382, ASFAK(S)VIGTP); WNK3-total antibody (S156C, second bleed; raised against residues 1142–1461 of human WNK3); SPAK-total antibody (S551D, third bleed; raised against full-length GST-tagged human SPAK protein); OSR1-total antibody (S850C, second bleed; raised against RSAHLPQPAGQMPTQPAQVSLR, residues 389–408 of mouse OSR1); SPAK/OSR1 (S-motif) phosphoS373/S325 antibody (S670B, second bleed; raised against 367–379 of human SPAK, RRVPGS(S)GHLHKT, which is highly similar to residues 319–331 of human OSR1 in which the sequence is RRVPGS(S)GRLHKT). KCC2 total antibody (residues 932–1043 of rat KCC2) was purchased from NeuroMab. The anti-β-Tubulin III (neuronal) antibody (T8578) was purchased from Sigma-Aldrich. Secondary antibodies coupled to horseradish peroxidase used for immunoblotting were obtained from Pierce. IgG used in control immunoprecipitation experiments was affinity-purified from pre-immune serum using Protein G-Sepharose.

**Gramicidin perforated patch recordings.** Experiments were performed on cells after 20–24 days in culture. Neurons were selected based on their GFP fluorescence indicating co-expression of GFP with glycine receptors. Recordings were made

using an Axopatch 200B amplifier (Molecular Devices), filtered at 5 kHz and digitized at 20 kHz. Neurons were perfused with extracellular solution (in mM) 125 NaCl, 20 D-glucose, 10 HEPES, 4 MgCl₂, 2 KCl, 1 CaCl₂, pH 7.4 containing TTX (1 μM), kynurenate (1 mM) and R,S-MCPG (500 μM) in a recording chamber (BadController V; Luigs and Neumann) at 31 °C mounted on an upright microscope (BX51WI; Olympus). Gramicidin perforated (50 μg/ml) patch recordings were performed using glass pipettes with a standard internal solution (in mM) 120 K-gluconate, 10 KCl, 10 HEPES, 0.1 EGTA, 4 MgATP 2H20, 0.4 Na₃GTP 2H₂0, pH 7.4. Recordings were made in voltage-clamp mode using an Axopatch 200B amplifier (Molecular Devices) and filtered at 2 kHz and digitized at 20 kHz. The membrane potential was held at −65 mV and depolarizing voltage steps were applied from −30 to +30 mV for 3 s, during which time either 100 μm glycine was puffed onto the soma using a Picospritzer (Parker Hannifin) or Rubi-GABA (15 μM; Tocris) was photolyzed at the soma using a digital modulated diode laser beam at 405 nm (Omicron Deepstar; Photon Lines) as previously described[3]. Glycinergic and GABAergic post-synaptic currents were measured and a linear regression of the current-voltage relationship was used to determine $E_{glycine}$ or $E_{GABA}$, respectively. Access and input resistance were monitored using a −5 mV step as previously described[3]. Voltages were corrected for the liquid junction potential (−16.2 mV) and access resistance was compensated offline.

**Whole-cell patch clamp recordings.** Neurons were recorded at 31 °C under superfusion with artificial cerebrospinal fluid containing (in mM) 125 NaCl, 20 D-glucose, 10 HEPES, 4 MgCl₂, 2 KCl, 1 CaCl₂ (pH = 7.4). Whole-cell recordings measuring GABABR currents were made with glass pipettes with the following internal solution (in mM): 110 K-methylsulfonate, 20 KCl, 10 HEPES, 10 EGTA, 4 MgATP, 0.4 Na₃GTP, 10 Na phosphocreatine, 1.8 MgCl₂. Recordings were made in the presence of TTX (1 μM), kynurenate (1 mM), and R,S-MCGP (500 μM), and bicuculline (20 μM). In current-clamp configuration the resting membrane potential (Vm) was monitored for 5 min to ensure a stable baseline, followed by the addition of bath applied baclofen (20 μM). Once Vm reached a steady-state in the baclofen treatment, the drug was either washed out or the GABABR antagonist CGP52132 (20 μM) was added to the bath. The Vm was monitored until it reached a steady-state. Whole-cell patch clamp recordings were performed with borosilicate glass micropipettes filled with either (in mM) 135 CsCl, 10 HEPES, 10 EGTA, 4 MgATP and 0.4 Na₃GTP (pH = 7.4) for recording tonic, GABAAR-mediated currents or 105 CsMeSO₄, 10 CsCl, 10 HEPES, 10 EGTA, 4 MgATP, and 0.4 Na₃GTP for recording muscimol-evoked currents. Cells were held at −70 mV. GABAAR-mediated currents were recorded in the presence of TTX (1 μM), NBQX (10 μM) and D,L-APV (100 μM). Access and input resistance were regularly monitored with −5 mV voltage steps. All recordings were made using an Axopatch 200B amplifier (Molecular Devices), filtered at 2 kHz and digitized at 25 kHz. All the data were collected using the Clampex 10 program and analyzed using Clampfit 10 (Axon). Currents were analyzed offline using Clampfit software.

**In vivo pentylenetatrazole injection.** Adult (postnatal day 84-91) C57bl6 mice (all males from JanvierLabs) were injected subcutaneously with pentylenetatrazole (PTZ, 75 mg/kg, dissolved in saline) and recorded right after injection using video recordings. The procedure was made in accordance with the guidelines of the French Agriculture and Forestry Ministry for handling animals and with the agreement of the Comité National de Réflexion Ethique sur l'Expérimentation Animale (#4018). The sampling of animals, as well as the experimental procedure and analysis of the data were determined based on previous published work. The animals to be used for PTZ vs. Control conditions were randomly chosen from the batch of C57bl6 mice delivered from JanvierLabs. After 20-35 min of observation, animals were killed by cervical dislocation, brains were rapidly extracted on ice and the cortex and the hippocampus were dissected, frozen in liquid nitrogen and stored at −80 °C until use for biochemistry. The first seizures were observed after 6–7 min of injection and a second sequence of seizures and/or abnormal gait were detected after 14–15 min. Two out of three animals injected with 75 mg/kg of PTZ had tonic-clonic seizures with rigid paw extension followed by death and one animal showed only partial clonus (as defined using Racine's scale).

**Statistics.** Sampling corresponds to the number of quantum dots for SPT, number of cultures or animals for biochemistry, cells for ICC, chloride and calcium imaging, and electrophysiology experiments. Sample size selection for experiments was based on published experiments, pilot studies, as well as in-house expertise. All results were used for analysis except in few cases. For imaging experiments (chloride and calcium imaging, SPT, immunofluorescence), cells with signs of suffering (apparition of blobs, fragmented neurites) were discarded from the analysis. For PTZ-treatment in vivo, one animal with an incorrect injection site was excluded from analysis. Means are shown ± SEM, median values are indicated with their interquartile range (IQR, 25–75%). Means were compared using the non-parametric Mann–Whitney test (immunocytochemistry, dwell time comparison), paired t-test (calcium imaging), Wilcoxon rank-sum test or paired t-test when normality test passed (chloride imaging) or two-tailed Student's t-test (biochemistry and gramicidin-perforated patch clamp) using SigmaPlot 12.5 software (Systat Software). Diffusion coefficient and explored area values having non-normal

distributions, a non-parametric Kolmogorov–Smirnov test was used. Median values were compared using the Kolmogorov–Smirnov test under Matlab (The Mathworks, Natick, MA). Differences were considered significant for $p$-values less than 5% (*$p \leq 0.05$; **$p < 0.01$; ***$p < 0.001$; NS, not significant).

**Data availability**. The data that support the findings of this study are available from the corresponding author upon reasonable request.

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

## Acknowledgments

We thank J. Nabekura for kindly providing the original pEGFP-IRES-KCC2 full-length construct, D Choquet for the homer1c-GFP construct, A. Triller for gephyrin-mRFP and glycine receptor α1 subunit constructs, I. Medina for WNK1-KD, WNK1-CA, shRNA against rat WNK1 mRNA and a scrambled shRNA sequence. We also thank M. Mameli, J. Hadchouel and P. Blaesse for critical reading of the manuscript. We are also grateful to the Cell and Tissue Imaging Facility of Institut du Fer à Moulin (IFM). This work was supported in part by Inserm, Sorbonne Université-UPMC, as well as the Fondation pour la Recherche Médicale (Equipe FRM DEQ20140329539 to J.C.P.), the Human Frontier Science Program (RGP0022/2013 to J.C.P.) and the Fondation pour la Recherche sur le Cerveau (to S.L.). Equipment at the IFM was also supported by DIM NeRF from Région Ile-de-France and by the FRC/Rotary 'Espoir en tête'. M.H. was the recipient of a doctoral fellowship from the Université Pierre and Marie Curie, as well as from Bio-Psy Laboratory of excellence. K.T.K. is supported by the National Institutes of Health, the Simons Foundation, and the March of Dimes Foundation Basil O'Connor Award. The Poncer/Lévi lab is affiliated with the Paris School of Neuroscience (ENP) and the Bio-Psy Laboratory of excellence.

## Author contributions

S.L. and M.H. designed the research. M.H. and M.Re. performed the single-particle tracking experiments and analyzed the data. M.H. and S.L. performed immuno-fluorescence experiments and M.H., S.L. and M.Ru. analyzed the data. J.P. and J.C.P. performed and analyzed electrophysiological experiments. J.Z. and K.T.K. designed the biochemical experiments; J.Z. performed the experiments; J.Z. and K.T.K. analyzed the data. S.A.A. and J.C.P. designed chloride imaging experiments and S.A.A. performed and analyzed the data. F.G.-C. performed the epilepsy experiments. E.E. and M.Ru. prepared the hippocampal cultures and I.M. and M.Re. contributed to new reagents and analytical tools. M.H. and S.L. prepared the figures and wrote the paper.

## Additional information

**Competing interests:** The authors declare no competing financial interests.

