## [Peer Review File · Nature Communications]

Reviewers' comments:

Reviewer #1 (Remarks to the Author):

The present work is aimed to determine the role of GABAergic inhibition on KCC2 membrane dynamics, clustering, stability and function in mature hippocampal neurons in vitro. Using a combination of elegant approaches, they provide us with a plausible model whereby GABA-mediated inhibition confines KCC2 in the plasma membrane via Cl⁻ as a second messenger and WNK1-mediated phosphorylation of KCC2 at specific residues. As the authors also conclude, this mechanism may contribute to the rapid homeostatic control of synaptic inhibition.

Although purely in vitro, this is a very nice and important piece of work with straightforward reasoning and concise writing. Results are novel, important, clearly presented and well discussed. There are, nevertheless a few issues that need clarifications:

Major concerns:

Results section p.18 describes KCC2-Flag immunoreactivity experiments that are performed to determine if KCC2 clustering is affected by gabazine or muscimol application. Details about experiment can be found in Figure 7 legend: "Ctrl, n = 62; Gbz, n = 59; Musc, n = 58; 4 cultures." This seems to indicate that "n" correspond to the number of cluster, and that 4 cultures in total were used for this set of experiments, leading the reader to suppose that 2 immunoreactivity experiments were done for one condition, not knowing which is it, and that the other conditions were tested only once. If this is the case, this result would need to be confirmed by a greater number of experiments.

Results section p.20 describes imaging experiments in which authors try to determine if muscimol or gabazine treatment have an impact on spine head area. As for the previous point, Figure 9 legend for A, B, C indicate that 3 cultures were used for this set of experiments, meaning a sample of 1 per condition. If this is exact, this is absolutely not sufficient to draw any conclusion. Moreover, as detected changes are below the imaging resolution (less than 160 μm as it seems), sample size should be elevated.

Same remarks for figures 9C, D, E.

Minor points:

- For quantum dot-based single particle tracking experiments, tables in the supplementary material indicate n of neurons and n of cultures. In this work, is sample element "neuron" or "culture"? Authors should specify this their statistics section.

In their Results p.5, authors wrote "...using quantum dot-based single particle tracking (QD-SPT) technique in (DIV 21-24) hippocampal cultured neurons. At this stage, neurons exhibit a mature phenotype, in that KCC2 surface expression has reached its maximal level 3. " But information about KCC2 surface expression as a function of DIV does not exist in the cited work. Authors should be more specific on this point or correct their reference.

n of neurons and cultures is missing for Figure S1 C.

In their Results p.6, authors wrote "Quantitative analysis performed on the bulk population (extrasynaptic + synaptic) of trajectories revealed a slight but not significant increase in the

diffusion coefficient of KCC2 after muscimol (Figure 1B)." First, as a general comment, a variation that is not statistically significant should not be considered as else than the result of natural fluctuations inside the sample, therefore should not be mentioned as an "increase" or a "decrease", but rather described as "not significantly different". Second, reader may have a hard time anyway to see a decrease of this parameter in Figure 1B.

In figure 9 legend, authors wrote "Note the increase in spine head area after gabazine ($p=0.005$) but not muscimol ($p=0.978$) treatment." The term "after" may be misleading as it indicates that this is a "before-after" type of experiment, whereas these are separated ones. Therefore "after" should be replaced for example by "in the condition of".

In the Discussion section p.24, the authors wrote: "Activation of GABAergic synapses leads to a local change in $[Cl^-]_i$ " and cite ref 23. This is followed by "Changes in the local activity of GABAergic synapses could hence rapidly alter KCC2 membrane expression through WNK1-dependent KCC2 phosphorylation. GABAAR activation and a subsequent increase in KCC2 confinement would maintain KCC2-mediated Cl^- extrusion capacity and help to counteract significant Cl^- influx through opened GABAARs to permit subsequent hyperpolarizing responses to GABA." This paragraph may lead the reader to understand that a local regulation of $[Cl^-]_i$ may occur through the local regulation of KCC2 confinement. But ref 23 indicate indeed that "diffusion is responsible for redistributing (and thus mitigating) transient, local changes in Cl^- load, while KCC2 level controls the steady-state balance of Cl^- influx and efflux." The present work is based on bath application-induced changes, meaning large and global effects on $[Cl^-]_i$. This point should be discussed more accurately in order to avoid misinterpretation by the reader.

It may be possible to improve Figure 10 clarity. On the left, arrow leading from KCC2 to increase of $[Cl^-]$ seems to indicate that this occurs through a decrease of $[Cl^-]$, which is obviously not the case (same kind of problem on the right panel). Instead, the arrow from KCC2 of the left panel could point on the decrease of $[Cl^-]$ on the right panel (below GABAAR), and the arrow from KCC2-P could point on the increase of $[Cl^-]$ on the left panel (below GABAAR).

In their Methods section p.27, authors wrote "KCC2-Flag constructs with Threonine residues 906 and 1007 mutated to glutamate (T906/1007E) or alanine (T906/1007A) were generated by Genscript (Piscataway, USA)." There is no problem buying constructs from Genescript, but is it correct to write in the results section p.17 "we generated KCC2 constructs with T906 and T1007 residues mutated to either glutamate (E) (T906/T1007E) or alanine (T906/T1007A)"?

In the References section, ref. 4 is incomplete.

Reviewer #2 (Remarks to the Author):

The study touches on an intriguing and potentially very important topic, whether KCC2 surface trafficking is affected by GABAA receptor activation itself and change in $[Cl^-]_i$, providing for an intrinsic homeostatic mechanism between the strength of Cl^- currents and regulation of Cl^- gradient to maintain inhibition constant. The authors present an important body of data. The data build on recent studies showing regulation of KCC2 membrane dynamics as well as KCC2 regulation by WNK kinases. The novelty here would be evidence of involvement of $[Cl^-]_i$ itself in the regulatory loop. I find that the data, as presented, fall short of meeting the standards for Nat. Commun. however.

Major comments:

The evidence of the contribution of $[Cl^-]_i$ is problematic in my sense. The evidence relies very heavily on use of GABAA agonists and antagonists to manipulate $[Cl^-]_i$, but there is no direct measurement of $[Cl^-]_i$. Agonist activation of GABAAR may not significantly affect $[Cl^-]_i$ for example if

EGABA quickly reaches V_m during prolonged activation in unclamped conditions. This is likely the case since the authors used HEPES buffered conditions, where the HCO_3^- contribution to EGABA is removed. This could explain a lot of the negative results with agonist applications. Thus the degree to which gabazine and muscimol affect $[\text{Cl}]_i$ is ill defined, and likely variable between conditions, especially if some manipulations affect GABAAR activity directly. The only experiment attempting to measure $[\text{Cl}]_i$ (Fig. 8) using EGly measurements is problematic and raises several concerns: throughout, the authors refer to the fact that gabazine lowers $[\text{Cl}]_i$ by decreasing GABAAR activity, but in the experiment shown in Fig. 8 they show an increase in $[\text{Cl}]_i$ (depolarized EGly); they argue that this is due to decreased KCC2 activity, but then their argument is circular since they first argue that the decrease in KCC2 is due to a decrease in $[\text{Cl}]_i$. They can't have it both ways. I believe two crucial experimental conditions are required for the study to be convincing: 1) imaging data confirming that several of the effects the authors see are related to changes in $[\text{Cl}]_i$; 2) independent manipulation of $[\text{Cl}]_i$ to show an effect on KCC2 membrane dynamics (e.g., manipulation of $[\text{Cl}]_i$ via light-activated NpHR for example).

Quantum Dot tracking: The data set presented does not allow one to evaluate how easily the KCC2-QD could be distinguished and quantified on single neurons. Representative widefield image with the 60X objective showing cells and the tracked QD could help understand/visualize the data sets.

What is the error on localization value for a single QD? Did the author compute this number?

What is the minimum number of frames for a single receptor trajectory? This number is important in the MSD curve fitting to obtain a realistic diffusion coefficient value.

To calculate the diffusion coefficient the authors used equation $\text{MSD}(t) = 4Dt + b$. Since b reflects the average spot localization accuracy of a trajectory, what is the average value for b ?

Statistical analysis: I have concerns with the approach used for the analysis. The authors rely heavily on distribution analysis compiling all of the QDs across cells using Kolmogorov-Smirnov tests. This yields a test that detects very small changes in distributions because the N is so large. Yet when one compares mean values (e.g. Fig. 4B vs 4D) very similar bar graphs yield very distinct levels of significance ($p=0.228$ vs. $p=0.032$; also e.g. Fig. 5F vs 5H). An additional statistical comparison using average values per cell and cell numbers as N would yield more convincing data that the statistics are not picking up differences from anecdotal comparisons of distributions.

In fact, it is difficult to follow the data tabulated in the supplementary figures with that in the main paper. Cross reference to figure number and letters would help. It is also troublesome to see that in many cases cell numbers vary but QD numbers are often rigorously the same. Did the authors select QDs? It is also not clear why the N used for measurement of Explored Area is twice that of the coefficient of diffusion. Aren't both values obtained from fit of MSD trajectories? Please clarify.

The display of the median boxplot 25 ± 75 , does not allow the reader to visualize full the distribution of the data. Adding histograms, cumulative frequencies or even showing the median box plot as $5 \pm 95\%$ would help this case.

GABAAR- $\alpha 5$ involvement: the authors should test for the impact of L-655,708 alone on the KCC2 membrane diffusion.

Do the authors have positive controls showing that Baclofen or CGP 52432 had an effect on the cells? Especially an effect on membrane potential?

Imaging data confirming that gabazine had no effect on $[\text{Ca}^{2+}]_i$, even in the presence of TTX-KYN-MCPG- Cd^{2+} , would strengthen the argument of the authors.

The data on closantel (Fig. 5F) is not conclusive since in the authors do not have a positive control: in the control condition presented, the gabazine application did not increase significantly the diffusion coefficient of KCC2. Thus the conclusion on the implication of SPAK/OSR1 in that experiment is unwarranted.

The authors refer to the fact that WNK3 kinase is low in adult neurons (Discussion), but the results shown in Fig 5C-H appear to reveal significant WNK3 expression in hippocampus (albeit 9 fold lower than WNK1). Can the authors discard any significant contribution of WNK3 to the results?

Recent studies (e.g., ref 18) have implicated involvement of WNK1 in pathological conditions which reflect long-term effects. But in the present study, the authors argue for very rapid effects (30 min treatment with gbz). This should be discussed. Are the two phenomena comparable?

From what I gather, in Fig 6. KCC2 was immunoprecipitated using KCC2 pT906 antibodies and immunoblotted with conventional KCC2 antibodies. Yet the KCC2 pT906 antibodies were raised against a KCC3A epitope that is highly homologous to other KCCs sequence (ref 27). Since other KCCs are expressed in hippocampal neurons it is possible that some of the effects observed reflect altered expression of other KCCs (e.g., KCC3A, which is particularly abundant in the CNS). These may have competed with KCC2 binding to protein-G-sepharose-coupled KCC2 pT906 antibodies. Additional control are necessary to rule out contribution of other KCCs (e.g., KCC2 pT906/1007 IP; quantitative Immunoblotting for KCC1, 3, and 4).

In Fig 6B, the authors must also provide results with Gbz + shWNK1 and closantel to conclusively show that the phosphorylation state of KCC2 T906/T1007 is really WNK1 and SPAK/OSR1-dependent.

It is not clear why the western blot samples were heated for 1h at 37{degree sign}C? This is not a conventional approach to remove proteins from beads and denature proteins. The more conventional approach involves heating for ~5 min at ~95{degree sign}C, especially for KCC proteins. The method used here may explain why more oligomers are observed than monomers in Fig 6A and 7E. To show results in denaturing conditions, monomers must be in higher proportions than oligomers. The approach used may thus confound the interpretation of the results.

In Fig 7E, is the molecular weight ladder properly shown? KCC2 monomers have a MW of ~140 kDa. Monomers in the panel are situated closer to the 100 kDa markers than the 150 kDa ones.

In Fig 7 the authors must provide FLAG surface staining and cell surface biotinylation results with Gbz + shWNK1 and closantel, again to convincingly show that KCC2 membrane clustering and stability are WNK1 and SPAK/OSR1-dependent.

The same comment applies for Fig 9 regarding Gbz + shWNK1 and closantel conditions.

Minor points:

Fig 2A and B labeling a bit confusing. I suggest changing the color (e.g. grey) for the "2 μ M GABA + L655,708" condition.

Fig 5F-G. Why are the results for Median EA not shown here?

In Fig 6F one cannot see whether the results are significantly different for T906/1007 and T906/1007E (as in the experiment shown in Fig 6E). They appear very different in 6E and 6F.

Fig 7 B-D and F-G. One could improve readability by using color shading for the Musc condition.

Page 19 line 2: replace "(Figure 7D)" by "(Figure 7C-D)".

Reviewer #3 (Remarks to the Author):

In this paper the authors provide evidence for GABA-A receptor mediated regulation of KCC2 abundance and clustering via intracellular Cl⁻ in cultured hippocampal neurons. The data presented and the conclusions drawn are novel and present a model for homeostatic regulation that is interesting and adds to the field. I have several comments that should be addressed and that could strengthen the project.

Major:

This is entirely an in vitro study. The major effect is with GABA-A receptor inhibition rather than activation. Is there evidence for this mechanism being important for endogenous GABA signaling in the intact nervous system?

The authors do not report n values for any of the results. Most comparisons are analyzed with t-tests or Mann-Whitney tests. Data in Figs 6 and 7, however, seem to involve multiple comparisons. For example, T906/1007A and T906/1007E conditions in 6A and B appear to be the same-that being the case why were they not compared for all conditions using a Kuskal-Wallis test? Same is true for WNK1-KD data in Figs. 5 G and H. Also there were no statistical analyses of the time-averaged MSD results in Figs. 1 and 4.

The shRNA experiment lacks a control for the vector (with an irrelevant shRNA sequence) and either a second shRNA or a cDNA rescue to control for off-target effects.

Central to the model is the modulation of [Cl⁻]_i, which was indirectly measured by a depolarizing shift in transfected glycine receptor reversal potential induced by gabazine. A direct measure using a fluorescent indicator would strengthen the argument, as well as testing some other pharmacological manipulations in which [Cl⁻]_i was assumed to be altered, such as muscimol, closantel, and VU 0240551 .

Minor:

In many places the words "compared to" are used but the comparison described differences in two conditions that are similar. When describing differences between otherwise Reviewers' comments:

Reviewer #1 (Remarks to the Author):

The present work is aimed to determine the role of GABAergic inhibition on KCC2 membrane dynamics, clustering, stability and function in mature hippocampal neurons in vitro. Using a combination of elegant approaches, they provide us with a plausible model whereby by GABAA-mediated inhibition confines KCC2 in the plasma membrane via Cl⁻ as a second messenger and WNK1-mediated phosphorylation of KCC2 at specific residues. As the authors also conclude, this mechanism may contribute to the rapid homeostatic control of synaptic inhibition.

Although purely in vitro, this is a very nice and important piece of work with straightforward reasoning and concise writing. Results are novel, important, clearly presented and well discussed. There are, nevertheless a few issues that need clarifications:

Major concerns:

Results section p.18 describes KCC2-Flag immunoreactivity experiments that are performed to determine if KCC2 clustering is affected by gabazine or muscimol application. Details about experiment can be found in Figure 7 legend: "Ctrl, n = 62; Gbz, n = 59; Musc, n = 58; 4 cultures." This seems to indicate that "n" correspond to the number of cluster, and that 4 cultures in total were used for this set of experiments, leading the reader to suppose that 2 immunoreactivity experiments were done for one condition, not knowing which is it, and that the other conditions were tested only once. If this is the case, this result would need to be confirmed by a greater number of experiments.

Results section p.20 describes imaging experiments in which authors try to determine if muscimol or gabazine treatment have an impact on spine head area. As for the previous point, Figure 9 legend for A, B, C indicate that 3 cultures were used for this set of experiments, meaning a sample of 1 per condition. If this is exact, this is absolutely not sufficient to draw any conclusion. Moreover, as detected changes are below the imaging resolution (less than 160 μm as it seems), sample size should be elevated.

Same remarks for figures 9C, D, E.

Minor points:

- For quantum dot-based single particle tracking experiments, tables in the supplementary material indicate n of neurons and n of cultures. In this work, is sample element "neuron" or "culture"? Authors should specify this their statistics section.

In their Results p.5, authors wrote "...using quantum dot-based single particle tracking (QD-SPT) technique in (DIV 21-24) hippocampal cultured neurons. At this stage, neurons exhibit a mature phenotype, in that KCC2 surface expression has reached its maximal level 3. " But information about KCC2 surface expression as a function of DIV does not exist in the cited work. Authors should be more specific on this point or correct their reference.

n of neurons and cultures is missing for Figure S1 C.

In their Results p.6, authors wrote "Quantitative analysis performed on the bulk population (extrasynaptic + synaptic) of trajectories revealed a slight but not significant increase in the diffusion coefficient of KCC2 after muscimol (Figure 1B)." First, as a general comment, a variation that is not statistically significant should not be considered as else than the result of natural fluctuations inside the sample, therefore should not be mentioned as an "increase" or a "decrease", but rather described as "not significantly different". Second, reader may have a hard time anyway to see a decrease of this parameter in Figure 1B.

In figure 9 legend, authors wrote "Note the increase in spine head area after gabazine ($p=0.005$) but not muscimol ($p=0.978$) treatment." The term "after" may be misleading as it indicates that this is a "before-after" type of experiment, whereas these are separated ones. Therefore "after" should be replaced for example by "in the condition of".

In the Discussion section p.24, the authors wrote: "Activation of GABAergic synapses leads to a local change in $[\text{Cl}^-]_i$ " and cite ref 23. This is followed by "Changes in the local activity of GABAergic synapses could hence rapidly alter KCC2 membrane expression through WNK1-dependent KCC2 phosphorylation. GABAAR activation and a subsequent increase in KCC2 confinement would maintain KCC2-mediated Cl^- extrusion capacity and help to counteract significant Cl^- influx through opened GABAARs to permit subsequent hyperpolarizing responses to GABA." This paragraph may lead the reader to understand that a local regulation of $[\text{Cl}^-]_i$ may

occur through the local regulation of KCC2 confinement. But ref 23 indicate indeed that "diffusion is responsible for redistributing (and thus mitigating) transient, local changes in Cl⁻ load, while KCC2 level controls the steady-state balance of Cl⁻ influx and efflux." The present work is based on bath application-induced changes, meaning large and global effects on [Cl⁻]_i. This point should be discussed more accurately in order to avoid misinterpretation by the reader.

It may be possible to improve Figure 10 clarity. On the left, arrow leading from KCC2 to increase of [Cl⁻] seems to indicate that this occurs through a decrease of [Cl⁻], which is obviously not the case (same kind of problem on the right panel). Instead, the arrow from KCC2 of the left panel could point on the decrease of [Cl⁻] on the right panel (below GABAAR), and the arrow from KCC2-P could point on the increase of [Cl⁻] on the left panel (below GABAAR).

In their Methods section p.27, authors wrote "KCC2-Flag constructs with Threonine residues 906 and 1007 mutated to glutamate (T906/1007E) or alanine (T906/1007A) were generated by Genscript (Piscataway, USA)." There is no problem buying constructs from Genescript, but is it correct to write in the results section p.17 "we generated KCC2 constructs with T906 and T1007 residues mutated to either glutamate (E) (T906/T1007E) or alanine (T906/T1007A)"?

In the References section, ref. 4 is incomplete.

Reviewer #2 (Remarks to the Author):

The study touches on an intriguing and potentially very important topic, whether KCC2 surface trafficking is affected by GABA_A receptor activation itself and change in [Cl⁻]_i, providing for an intrinsic homeostatic mechanism between the strength of Cl⁻ currents and regulation of Cl⁻ gradient to maintain inhibition constant. The authors present an important body of data. The data build on recent studies showing regulation of KCC2 membrane dynamics as well as KCC2 regulation by WNK kinases. The novelty here would be evidence of involvement of [Cl⁻]_i itself in the regulatory loop. I find that the data, as presented, fall short of meeting the standards for Nat. Commun. however.

Major comments:

The evidence of the contribution of [Cl⁻]_i is problematic in my sense. The evidence relies very heavily on use of GABA_A agonists and antagonists to manipulate [Cl⁻]_i, but there is no direct measurement of [Cl⁻]_i. Agonist activation of GABA_AR may not significantly affect [Cl⁻]_i for example if EGABA quickly reaches V_m during prolonged activation in unclamped conditions. This is likely the case since the authors used HEPES buffered conditions, where the HCO₃⁻ contribution to EGABA is removed. This could explain a lot of the negative results with agonist applications. Thus the degree to which gabazine and muscimol affect [Cl⁻]_i is ill defined, and likely variable between conditions, especially if some manipulations affect GABA_AR activity directly. The only experiment attempting to measure [Cl⁻]_i (Fig. 8) using EGly measurements is problematic and raises several concerns: throughout, the authors refer to the fact that gabazine lowers [Cl⁻]_i by decreasing GABA_AR activity, but in the experiment shown in Fig. 8 they show an increase in [Cl⁻]_i (depolarized EGly); they argue that this is due to decreased KCC2 activity, but then their argument is circular since they first argue that the decrease in KCC2 is due to a decrease in [Cl⁻]_i. They can't have it both ways. I believe two crucial experimental conditions are required for the study to be convincing: 1) imaging data confirming that several of the effects the authors see are related to changes in [Cl⁻]_i; 2) independent manipulation of [Cl⁻]_i to show an effect on KCC2 membrane dynamics (e.g., manipulation of [Cl⁻]_i via light-activated NpHR for example).

Quantum Dot tracking: The data set presented does not allow one to evaluate how easily the KCC2-QD could be distinguished and quantified on single neurons. Representative widefield image with the 60X objective showing cells and the tracked QD could help understand/visualize the data

sets.

What is the error on localization value for a single QD? Did the author compute this number?

What is the minimum number of frames for a single receptor trajectory? This number is important in the MSD curve fitting to obtain a realistic diffusion coefficient value.

To calculate the diffusion coefficient the authors used equation $MSD(t) = 4Dt + b$. Since b reflects the average spot localization accuracy of a trajectory, what is the average value for b ?

Statistical analysis: I have concerns with the approach used for the analysis. The authors rely heavily on distribution analysis compiling all of the QDs across cells using Kolmogorov-Smirnov tests. This yields a test that detects very small changes in distributions because the N is so large. Yet when one compares mean values (e.g. Fig. 4B vs 4D) very similar bar graphs yield very distinct levels of significance ($p=0.228$ vs. $p=0.032$; also e.g. Fig. 5F vs 5H). An additional statistical comparison using average values per cell and cell numbers as N would yield more convincing data that the statistics are not picking up differences from anecdotal comparisons of distributions.

In fact, it is difficult to follow the data tabulated in the supplementary figures with that in the main paper. Cross reference to figure number and letters would help. It is also troublesome to see that in many cases cell numbers vary but QD numbers are often rigorously the same. Did the authors select QDs? It is also not clear why the N used for measurement of Explored Area is twice that of the coefficient of diffusion. Aren't both values obtained from fit of MSD trajectories? Please clarify.

The display of the median boxplot 25 ± 75 , does not allow the reader to visualize full the distribution of the data. Adding histograms, cumulative frequencies or even showing the median box plot as $5 \pm 95\%$ would help this case.

GABAAR- $\alpha 5$ involvement: the authors should test for the impact of L-655,708 alone on the KCC2 membrane diffusion.

Do the authors have positive controls showing that Baclofen or CGP 52432 had an effect on the cells? Especially an effect on membrane potential?

Imaging data confirming that gabazine had no effect on $[Ca^{2+}]$, even in the presence of TTX-KYN-MCPG- Cd^{2+} , would strengthen the argument of the authors.

The data on closantel (Fig. 5F) is not conclusive since in the authors do not have a positive control: in the control condition presented, the gabazine application did not increase significantly the diffusion coefficient of KCC2. Thus the conclusion on the implication of SPAK/OSR1 in that experiment is unwarranted.

The authors refer to the fact that WNK3 kinase is low in adult neurons (Discussion), but the results shown in Fig 5C-H appear to reveal significant WNK3 expression in hippocampus (albeit 9 fold lower than WNK1). Can the authors discard any significant contribution of WNK3 to the results?

Recent studies (e.g., ref 18) have implicated involvement of WNK1 in pathological conditions which reflect long-term effects. But in the present study, the authors argue for very rapid effects (30 min treatment with gbz). This should be discussed. Are the two phenomena comparable?

From what I gather, in Fig 6. KCC2 was immunoprecipitated using KCC2 pT906 antibodies and immunoblotted with conventional KCC2 antibodies. Yet the KCC2 pT906 antibodies were raised against a KCC3A epitope that is highly homologous to other KCCs sequence (ref 27). Since other KCCs are expressed in hippocampal neurons it is possible that some of the effects observed reflect

altered expression of other KCCs (e.g., KCC3A, which is particularly abundant in the CNS). These may have competed with KCC2 binding to protein-G-sepharose-coupled KCC2 pT906 antibodies. Additional controls are necessary to rule out contribution of other KCCs (e.g., KCC2 pT906/1007 IP; quantitative Immunoblotting for KCC1, 3, and 4).

In Fig 6B, the authors must also provide results with Gbz + shWnk1 and closantel to conclusively show that the phosphorylation state of KCC2 T906/T1007 is really Wnk1 and SPAK/OSR1-dependent.

It is not clear why the western blot samples were heated for 1h at 37°C? This is not a conventional approach to remove proteins from beads and denature proteins. The more conventional approach involves heating for ~5 min at ~95°C, especially for KCC proteins. The method used here may explain why more oligomers are observed than monomers in Fig 6A and 7E. To show results in denaturing conditions, monomers must be in higher proportions than oligomers. The approach used may thus confound the interpretation of the results.

In Fig 7E, is the molecular weight ladder properly shown? KCC2 monomers have a MW of ~140 kDa. Monomers in the panel are situated closer to the 100 kDa markers than the 150 kDa ones.

In Fig 7 the authors must provide FLAG surface staining and cell surface biotinylation results with Gbz + shWnk1 and closantel, again to convincingly show that KCC2 membrane clustering and stability are Wnk1 and SPAK/OSR1-dependent.

The same comment applies for Fig 9 regarding Gbz + shWnk1 and closantel conditions.

Minor points:

Fig 2A and B labeling a bit confusing. I suggest changing the color (e.g. grey) for the "2μM GABA + L655,708" condition.

Fig 5F-G. Why are the results for Median EA not shown here?

In Fig 6F one cannot see whether the results are significantly different for T906/1007 and T906/1007E (as in the experiment shown in Fig 6E). They appear very different in 6E and 6F.

Fig 7 B-D and F-G. One could improve readability by using color shading for the Musc condition.

Page 19 line 2: replace "(Figure 7D)" by "(Figure 7C-D)".

Reviewer #3 (Remarks to the Author):

In this paper the authors provide evidence for GABA-A receptor mediated regulation of KCC2 abundance and clustering via intracellular Cl⁻ in cultured hippocampal neurons. The data presented and the conclusions drawn are novel and present a model for homeostatic regulation that is interesting and adds to the field. I have several comments that should be addressed and that could strengthen the project.

Major:

This is entirely an in vitro study. The major effect is with GABA-A receptor inhibition rather than activation. Is there evidence for this mechanism being important for endogenous GABA signaling in the intact nervous system?

The authors do not report n values for any of the results. Most comparisons are analyzed with t-tests or Mann-Whitney tests. Data in Figs 6 and 7, however, seem to involve multiple comparisons. For example, T906/1007A and T906/1007E conditions in 6A and B appear to be the same-that being the case why were they not compared for all conditions using a Kuskal-Wallis test? Same is true for WNK1-KD data in Figs. 5 G and H. Also there were no statistical analyses of the time-averaged MSD results in Figs. 1 and 4.

The shRNA experiment lacks a control for the vector (with an irrelevant shRNA sequence) and either a second shRNA or a cDNA rescue to control for off-target effects.

Central to the model is the modulation of $[Cl^-]_i$, which was indirectly measured by a depolarizing shift in transfected glycine receptor reversal potential induced by gabazine. A direct measure using a fluorescent indicator would strengthen the argument, as well as testing some other pharmacological manipulations in which $[Cl^-]_i$ was assumed to be altered, such as muscimol, closantel, and VU 0240551 .

Minor:

In many places the words "compared to" are used but the comparison described differences in two conditions that are similar. When describing differences between otherwise

We thank our three referees for their helpful comments and suggestions. We are pleased that the referees acknowledged the importance and originality of our data, which uncovers a novel mechanism of KCC2 regulation by GABA_A receptor-mediated transmission. We have performed several new experiments and significantly modified our text to address all issues raised by the three referees. We think the manuscript has been improved and several important issues clarified based on the referees' suggestions. We now hope it will now be judged suitable for publication in *Nature Communications*.

Reviewer #1 (Remarks to the Author):

The present work is aimed to determine the role of GABAergic inhibition on KCC2 membrane dynamics, clustering, stability and function in mature hippocampal neurons in vitro. Using a combination of elegant approaches, they provide us with a plausible model whereby by GABA-mediated inhibition confines KCC2 in the plasma membrane via Cl⁻ as a second messenger and WNK1-mediated phosphorylation of KCC2 at specific residues. As the authors also conclude, this mechanism may contribute to the rapid homeostatic control of synaptic inhibition. Although purely in vitro, this is a very nice and important piece of work with straightforward reasoning and concise writing. Results are novel, important, clearly presented and well discussed. There are, nevertheless a few issues that need clarifications:

We are thankful to the reviewer for his/her enthusiasm and recognition of the accomplished work. We have addressed below specific points that the reviewer asked us to clarify. We thank him/her for the remarks that helped to improve the manuscript.

Major concerns:

Results section p.18 describes KCC2-Flag immunoreactivity experiments that are performed to determine if KCC2 clustering is affected by gabazine or muscimol application. Details about experiment can be found in Figure 7 legend: "Ctrl, n = 62; Gbz, n = 59; Musc, n = 58; 4 cultures." This seems to indicate that "n" correspond to the number of cluster, and that 4 cultures in total were used for this set of experiments, leading the reader to suppose that 2 immunoreactivity experiments were done for one condition, not knowing which is it, and that the other conditions were tested only once. If this is the case, this result would need to be confirmed by a greater number of experiments.

We believe the number of observations and experimental replicates are adequate in our ICC experiments. We apologize that this was not clearly explained in the legends. We have now clarified this point. In the legend: "Ctrl, n = 62; Gbz, n = 59; Musc, n = 58; 4 cultures", n corresponds to the total number of cells analyzed. Since ~100 clusters per cell were analyzed, this means that a minimum of 58*100= 5 800 clusters were taken into account for quantification. We have now clarified this point in the legend as follow: *Ctrl n= 77 cells, Gbz n= 83 cells ...; 4 cultures*. For a better understanding, the following sentence was added to the methods sub-section "Fluorescence image acquisition and analysis" : *For each culture, we analyzed ~10 cells per experimental condition and ~100 clusters or ~15 spines per cell (page 46)*.

Results section p.20 describes imaging experiments in which authors try to determine if muscimol or gabazine treatment have an impact on spine head area. As for the previous point, Figure 9 legend for A, B, C indicate that 3 cultures were used for this set of experiments, meaning a sample of 1 per condition. If this is exact, this is absolutely not sufficient to draw any conclusion. Moreover, as detected changes are below the imaging resolution (less than 160 μm as it seems), sample size should be elevated.

We apologize for this misunderstanding. We analyzed on average 10 cells per experimental condition and 15 spines per cell. In the legend of Figure 9 A-C: "Ctrl, n = 41; Gbz, n = 36; Musc, n = 39; 3 cultures", n corresponds to the total number of cells quantified. We have now clarified the legend by adding the number of spines quantified as follow: *Ctrl n= 1301 spines; Gbz n= 1082 spines; Musc n= 1014 spines, 3 cultures.*

Same remarks for figures 9C, D, E.

We have now clarified this point: *T906/T1007 n= 1301 spines, T906/T1007A n= 1224 spines, T906/T1007E n= 983 spines, T906/T1007A+Gbz n= 1093 spines, 3 cultures*

Minor points:

- For quantum dot-based single particle tracking experiments, tables in the supplementary material indicate n of neurons and n of cultures. In this work, is sample element "neuron" or "culture"? Authors should specify this their statistics section.

In all figures, n refers to the number of QDs, not the number of cells or neurons. To take into account the wide range of diffusion behaviors of individual molecules, it is usual in the Single Particle Tracking field to represent diffusion coefficients and explored areas as cumulative probabilities or median values +/- IQR instead of means +/- SEM. As asked by the reviewer, the following sentence has been added to the statistics section: *Sampling corresponds to the number of quantum dots for SPT, number of cultures or animals for biochemistry, cells for ICC, chloride and calcium imaging, and electrophysiology experiments. Sample size selection for experiments was based on published experiments, pilot studies as well as in-house expertise.*

In their Results p.5, authors wrote "...using quantum dot-based single particle tracking (QD-SPT) technique in (DIV 21-24) hippocampal cultured neurons. At this stage, neurons exhibit a mature phenotype, in that KCC2 surface expression has reached its maximal level 3. " But information about KCC2 surface expression as a function of DIV does not exist in the cited work. Authors should be more specific on this point or correct their reference.

The reviewer is correct. We had removed this piece of data from our PNAS paper. Below are our unpublished results showing the pattern of expression of KCC2 in hippocampal cultured neurons. We have therefore removed this reference and modified the sentence as follow: *At this stage, GABAAR-mediated responses are hyperpolarizing and inhibitory as revealed by EGABA measurements with gramicidin-perforated patch-clamp and local photolysis of caged GABA (Figure S1).*

Developmental expression profile of KCC2 in cultured hippocampal neurons

n of neurons and cultures is missing for Figure S1 C.

Done.

In their Results p.6, authors wrote "Quantitative analysis performed on the bulk population (extrasynaptic + synaptic) of trajectories revealed a slight but not significant increase in the diffusion coefficient of KCC2 after muscimol (Figure 1B)." First, as a general comment, a variation that is not statistically significant should not be considered as else than the result of natural fluctuations inside the sample, therefore should not be mentioned as an "increase" or a "decrease", but rather described as "not significantly different". Second, reader may have a hard time anyway to see a decrease of this parameter in Figure 1B.

We agree. The text was changed as follows: *Quantitative analysis performed on the bulk population (extrasynaptic + synaptic) of trajectories revealed no significant effect on the diffusion coefficient of KCC2 after muscimol (Figure 1B).*

In figure 9 legend, authors wrote "Note the increase in spine head area after gabazine ($p=0.005$) but not muscimol ($p=0.978$) treatment." The term "after" may be misleading as it indicates that this is a "before-after" type of experiment, whereas these are separated ones. Therefore "after" should be replaced for example by "in the condition of".

The text was modified as recommended by the reviewer and now reads: *Ctrl vs Gbz KS test $p=0.005$, Ctrl vs Musc KS test $p=0.978$.*

In the Discussion section p.24, the authors wrote: "Activation of GABAergic synapses leads to a local change in $[Cl^-]_i$ " and cite ref 23. This is followed by "Changes in the local activity of GABAergic synapses could hence rapidly alter KCC2 membrane expression through WNK1-dependent KCC2 phosphorylation. GABAAR activation and a subsequent increase in KCC2 confinement would maintain KCC2-mediated Cl^- extrusion capacity and help to counteract significant Cl^- influx through opened GABAARs to permit subsequent hyperpolarizing responses to GABA." This paragraph may lead the reader to understand that a local regulation of $[Cl^-]_i$ may occur through the local regulation of KCC2 confinement. But ref 23 indicate indeed that "diffusion is responsible for redistributing (and thus mitigating) transient, local

changes in Cl⁻ load, while KCC2 level controls the steady-state balance of Cl⁻ influx and efflux." The present work is based on bath application-induced changes, meaning large and global effects on [Cl⁻]_i. This point should be discussed more accurately in order to avoid misinterpretation by the reader.

Indeed the reviewer is correct about the fact that the computational data in ref 23 does not support our conclusion about a significant impact of local KCC2 confinement counteracting Cl⁻ influx at single synapses. Chloride measurements with highly sensitive chloride sensors may help answering this question more accurately. To not mislead the reader we changed our discussion to: "*Activation of GABAergic synapses leads to changes in [Cl⁻]_i³⁹. Changes in the activity of GABAergic synapses could hence rapidly alter KCC2 membrane expression through WNK1-dependent KCC2 phosphorylation. Changes in GABAAR activation and a subsequent increase in KCC2 confinement would maintain KCC2-mediated Cl⁻ extrusion and help counteract Cl⁻ influx through GABAARs.*" (page 35)

It may be possible to improve Figure 10 clarity. On the left, arrow leading from KCC2 to increase of [Cl⁻] seems to indicate that this occurs through a decrease of [Cl⁻], which is obviously not the case (same kind of problem on the right panel). Instead, the arrow from KCC2 of the left panel could point on the decrease of [Cl⁻] on the right panel (below GABAAR), and the arrow from KCC2-P could point on the increase of [Cl⁻] on the left panel (below GABAAR).

The figure has been changed.

In their Methods section p.27, authors wrote "KCC2-Flag constructs with Threonine residues 906 and 1007 mutated to glutamate (T906/1007E) or alanine (T906/1007A) were generated by Genscript (Piscataway, USA)." There is no problem buying constructs from Genescript, but is it correct to write in the results section p.17 "we generated KCC2 constructs with T906 and T1007 residues mutated to either glutamate (E) (T906/T1007E) or alanine (T906/T1007A)"?

As recommended by the reviewer, the following sentence was changed as follows: "*To test the involvement of KCC2 T906/T1007 phosphorylation in the gabazine-induced regulation of KCC2 diffusion, we expressed KCC2 constructs harboring mutation of T906 and T1007 to either glutamate (E) (T906/T1007E) or alanine (T906/T1007A) to mimic KCC2 T906 and T1007 phosphorylated or dephosphorylated states, respectively 17.*" (page 22)

In the References section, ref. 4 is incomplete.

We thank the reviewer for his/her vigilance. The reference has been corrected.

Reviewer #2 (Remarks to the Author):

The study touches on an intriguing and potentially very important topic, whether KCC2 surface trafficking is affected by GABAA receptor activation itself and change in [Cl⁻]_i, providing for an intrinsic homeostatic mechanism between the strength of Cl⁻ currents and regulation of Cl⁻ gradient to maintain inhibition constant. The authors present an important body of data. The data build on recent studies showing regulation of KCC2 membrane dynamics as well as KCC2

regulation by WNK kinases. The novelty here would be **evidence of involvement of $[Cl^-]_i$ itself in the regulatory loop**. I find that the data, as presented, fall short of meeting the standards for Nat. Commun. however.

Major comments:

The evidence of the contribution of $[Cl^-]_i$ is problematic in my sense. The evidence relies very heavily on use of GABA_A agonists and antagonists to manipulate $[Cl^-]_i$, but **there is no direct measurement of $[Cl^-]_i$** . **Agonist activation of GABA_AR may not significantly affect $[Cl^-]_i$** for example if EGABA quickly reaches V_m during prolonged activation in unclamped conditions. This is likely the case since the authors used **HEPES buffered conditions**, where the HCO₃ contribution to EGABA is removed. This could explain a lot of the negative results with agonist applications. Thus **the degree to which gabazine and muscimol affect $[Cl^-]_i$ is ill defined, and likely variable between conditions, especially if some manipulations affect GABA_AR activity directly**.

We agree with the referee that this information is critical to interpreting our data. Therefore, we have performed additional experiments using FRET imaging of SuperClomeleon chloride probe in hippocampal neurons to test whether our pharmacological manipulations produced the expected changes in $[Cl^-]_i$. Our results show that both muscimol and VU0240551 significantly increase $[Cl^-]_i$ in our conditions whereas extracellular chloride depletion decreased it. Gabazine however add no detectable effect on SuperClomeleon FRET ratio, likely owing to its lack of sensitivity in the 0-15 mM range in our conditions (Figure 4A).

We were particularly careful in performing chloride imaging in the same imaging medium that was used for SPT experiments so that results could be directly compared. The modest effect of muscimol and VU and now halorhodopsin (see below) on KCC2 diffusion and clustering cannot be attributed to the presence of HEPES in the medium as similar conclusions were obtained from biochemical assays where drugs were directly added to the culture medium which is bicarbonate-buffered.

The only experiment attempting to measure $[Cl^-]_i$ (Fig. 8) using EGly measurements is problematic and raises several concerns: throughout, the authors refer to the fact that gabazine lowers $[Cl^-]_i$ by decreasing GABA_AR activity, but in the experiment shown in Fig. 8 they show an increase in $[Cl^-]_i$ (depolarized EGly); they argue that this is due to decreased KCC2 activity, but then their argument is circular since they first argue that the decrease in KCC2 is due to a decrease in $[Cl^-]_i$. They can't have it both ways.

We appreciate the referee's concern. However, we believe that our data can be easily interpreted since changes in $[Cl^-]_i$ upon gabazine application occur on different time scales. Thus, although SuperClomeleon imaging could not detect an immediate drop in $[Cl^-]_i$ expected upon gabazine application, we observed a delayed increase in $[Cl^-]_i$ that is consistent with our electrophysiological recordings. We propose that gabazine application leads to a transient drop in $[Cl^-]_i$ that in turn activates the WNK/SPAK/OSR signaling pathway leading to KCC2 phosphorylation in T906 and T1007. This effect is likely to persist well beyond the initial drop in $[Cl^-]_i$ as it will be reversed through the recruitment of PP1 phosphatase (Darman et al. J Biol Chem. 2001 276(37):34359-62; Gagnon and Delpire, J Biol Chem. 2010 ; 285(19):14115-21).

Persistent KCC2 inactivation by threonine phosphorylation would then be expected to translate into a rise in $[Cl^-]_i$, consistent with our electrophysiological and chloride imaging data. We have tried to clarify this point in our revised manuscript (page 35).

I believe two crucial experimental conditions are required for the study to be convincing: 1) imaging data confirming that several of the effects the authors see are related to changes in $[Cl^-]_i$; 2) independent manipulation of $[Cl^-]_i$ to show an effect on KCC2 membrane dynamics (e.g., manipulation of $[Cl^-]_i$ via light-activated NpHR for example).

Concerning point 1, we have now directly addressed this point by FRET imaging of the chloride sensor SuperClomeleon in hippocampal neurons (Figure 4A-B). As expected, muscimol and VU0240551 significantly increased $[Cl^-]_i$ while lowering $[Cl^-]_i$ by substituting extracellular Cl^- with methanesulfonate decreased $[Cl^-]_i$. These data have been discussed above. Note that closantel and shRNAs against WNK1 or WNK3 were not tested for effect on $[Cl^-]_i$, however, since they are targeting downstream chloride effectors.

Concerning point 2, the effects of $[Cl^-]_i$ manipulations independent of GABAAR activity were substantially addressed in our work (Figure 4). We raised $[Cl^-]_i$ by exposing neurons to a selective KCC2 inhibitor and lowered it by substituting extracellular Cl^- with methanesulfonate in the imaging medium. Our results clearly show that these manipulations, independent of GABAAR manipulations, reproduce the effect of GABAAR agonists and antagonists, respectively.

According to the reviewer's recommendations, we have now directly address the effect of rising $[Cl^-]_i$ with light-activated eNpHR on KCC2 diffusion behavior. We found that the photostimulation of halorhodopsin eNpHR significantly reduced the diffusion coefficient (by ~2-fold) and increased the confinement (by 2-3-fold) of KCC2 transporter. This was observed as soon as 10s after light exposure. This experiment clearly reveal that increasing $[Cl^-]_i$ confines the transporter in the plasma membrane. The results of this experiment have been added to Figure 4.

Collectively, our results provide strong evidence that intracellular Cl^- acts as a second messenger to rapidly modulate KCC2 diffusion downstream of changes in the activity of GABAARs.

Quantum Dot tracking: The data set presented does not allow one to evaluate how easily the KCC2-QD could be distinguished and quantified on single neurons. Representative widefield image with the 60X objective showing cells and the tracked QD could help understand/visualize the data sets.

Here is a representative 60X image (left) of a KCC2-Flag and GFP double-transfected neuron labeled live for QD-based single particle imaging for referee's appreciation. This image shows QD trajectories (red) reconstructed with MatLab software overlaid with GFP fluorescence (gray). One to two sub-regions of dendrites (boxed, and shown at higher magnification in right panel) were quantified per cell. Note that QDs are only detected on transfected cell (specific staining), reconstruction of trajectories with MatLab software is very efficient and QDs are sufficiently spaced from one another to prevent QD crossing. In cases of QD crossing, the trajectories are discarded from analysis, as indicated in the Methods section of our manuscript (page 42).

What is the error on localization value for a single QD? Did the author compute this number?

We indeed measured this number on our setup. We calculated the variance of the position of QDs immobilized (dried) on a coverslip on real time recordings acquired with the same parameters of acquisition (integration time of 30 ms over 1200 consecutive frames) than those used for SPT experiments. The QD pointing accuracy is ~20-30 nm depending on the type of lamp used, a value well below the explored areas measured here (at least 1 log difference). Therefore, experimental values are not compromised by QD pointing accuracy. The following sentence was added in the methods section: *"Depending on the type of lamp used for imaging, the QD pointing accuracy is ~20-30 nm, a value well below the measured explored areas (at least 1 log difference)."* (page 42)

What is the minimum number of frames for a single receptor trajectory? This number is important in the MSD curve fitting to obtain a realistic diffusion coefficient value.

The reviewer is correct. Calculation of the Diffusion coefficients (D) is done by fitting the MSD vs time plot (which is very noisy) for short trajectories. As D is calculated by fitting the time points 2 to 5 (without origin) of the MSD (Kusumi et al., Biophys J. 1993;65(5):2021-40), trajectories of less than 15 time points were not considered in order to increase the accuracy in D calculation. Furthermore, in contrast to organic dyes, the high photostability of QDs allow to reconstruct trajectories of 15-1200 points without any difficulty. The measurement of D is therefore more accurate when using QDs than organic dyes.

To calculate the diffusion coefficient the authors used equation $MSD(t) = 4Dt + b$. Since b reflects the average spot localization accuracy of a trajectory, what is the average value for b?

Knowing that the pointing accuracy is ~ 30 nm, b is $\sim 7 \times 10^{-4} \mu\text{m}^2$. See our answer above.

Statistical analysis: I have concerns with the approach used for the analysis. The authors rely heavily on distribution analysis compiling all of the QDs across cells using Kolmogorov-Smirnov tests. This yields a test that detects very small changes in distributions because the N is so large. Yet when one compares **mean** values (e.g. Fig. 4B vs 4D) very similar bar graphs yield very distinct levels of significance ($p=0.228$ vs. $p=0.032$; also e.g. Fig. 5F vs 5H).

The distributions of diffusion coefficient and explored area values are not normal. The mean values are therefore not appropriate to represent these variables. This is why we never show or compare mean values of diffusion coefficient or explored area when studying protein mobility in cell membranes. The graphs in Figure 4D and 4F (previously Figure 4B and 4D) are therefore not representing the mean values but the median values of diffusion coefficient or explored area + 25%-75% interquartile range (like in all other graphs). Accordingly, statistical comparison was always performed on distributions, not mean values.

An additional statistical comparison using average values per cell and cell numbers as N would yield more convincing data that the statistics are not picking up differences from anecdotal comparisons of distributions.

Transmembrane proteins undergo a wide variety of movement at the surface of neurons. It is therefore not accurate to compare mobility between cells since the number of trajectories per cell is too low to be representative of the wide range of diffusion behaviors. This is why we need to pool data from many cells (as explained in Choquet and Triller Neuron 2008, 59(3):359-74). Knowing that transmembrane proteins undergo a large range of movements, it is therefore more accurate to compare the distributions of diffusion coefficient and explored area values with the Kolmogorov-Smirnov test rather than comparing the median values with the Mann Whitney test. Indeed median values can change or not depending whether the mobility of only one subpopulation of molecules is altered.

In fact, it is difficult to follow the data tabulated in the supplementary figures with that in the main paper. Cross reference to figure number and letters would help.

This has been done.

It is also troublesome to see that in many cases cell numbers vary but QD numbers are often rigorously the same. Did the authors select QDs?

The numbers of QDs vary from one cell to another and depending on the experimental condition tested (since some conditions led to transporter internalization or not). The total number of trajectories is obtained only at the end of an experiment. To prevent giving too much weight to an experimental condition in particular (with larger numbers) and in order to compare the conditions between them within the same culture, we reduced the number of QDs in the condition with the largest n to fit the same number than the condition with the smallest n . For this, conditions with high n were sorted randomly. This indeed reduced the total number of QDs analyzed per experiment. Moreover, we always checked that the statistical difference measured on the whole population was going in the same direction than

the one with smallest n. A sentence explaining this procedure has been now added to the methods section (page 43).

It is also not clear why the N used for measurement of Explored Area is twice that of the coefficient of diffusion. Aren't both values obtained from fit of MSD trajectories? Please clarify.

The diffusion coefficient value is a fit on each trajectory. In contrast, the explored area of each trajectory does not come from a fitting procedure. It was defined as the MSD value of the trajectory at two different time intervals i.e. at 0.42 and 0.45 s (Renner et al., PLoS One. 2012;7(8):e43032). This is why the n of explored area is twice that of diffusion coefficient. We have now clarified this point in the methods section (page 42).

The display of the median boxplot 25+/- 75, does not allow the reader to visualize full the distribution of the data. Adding histograms, cumulative frequencies or even showing the median box plot as 5 +/- 95 % would help this case.

The cumulative distributions are a good way to show diffusion data when comparing two different conditions only. When comparing more than two conditions (which is the case in most of our experiments), these plots loose clarity. This is why we chose the median boxplot \pm 25%-75% IQR representation that allows comparison of many conditions at the same time and is a standard in the field. In any case, this is just a representation issue, as statistical differences are indeed estimated from the whole distribution using Kolmogorov-Smirnov tests.

GABAAR-alpha5 involvement: the authors should test for the impact of L-655,708 alone on the KCC2 membrane diffusion.

Done. These experiments show that addition of L-655,708 (50 μ M) in absence of exogenous GABA, had no effect on KCC2 diffusion. These results have been added to Figure 2A-B and Table S2. This results supports our conclusion that tonic GABAAR-dependent currents are undetectable in our cultured hippocampal neurons in the absence of GABA application (Figure S1), as reported earlier (Eugene et al. J Neurosci 2007). In contrast, addition of L-655,708 to GABA (2 μ M) increased KCC2 mobility as compared to GABA alone, indicating that tonic GABAAR-mediated signaling may affect KCC2 mobility.

Do the authors have positive controls showing that Baclofen or CGP 52432 had an effect on the cells? Especially an effect on membrane potential?

We had not initially performed this important control but now provide new experimental data to address this point (Figure S3). We found that bath application of baclofen (20 μ M) significantly hyperpolarized neurons by about 8mV in our experimental conditions, and this effect was both reversible and blocked by the GABABR antagonist CGP 52432 (20 μ M).

Imaging data confirming that gabazine had no effect on $[Ca^{2+}]_i$, even in the presence of TTX-KYN-MCPG-Cd²⁺, would strengthen the argument of the authors.

We agree with the referee this is also an important control. We have therefore performed additional experiments to test this point using Fluo4AM-based Ca^{2+} imaging in the presence

of TTX+kynurenate+MCPG+ Cd^{2+} and in absence or presence of gabazine. Gabazine application to neurons in these conditions had no effect on $[\text{Ca}^{2+}]_i$ in all cells analyzed (Figure 3A-B). In contrast, application of NMDA (50 μM) to neurons maintained in the absence of TTX and glutamate receptor antagonists rapidly increased $[\text{Ca}^{2+}]_i$ (Figure 3C-D). These results show that gabazine has no effect on $[\text{Ca}^{2+}]_i$ in neurons maintained in TTX+kynurenate+MCPG+ Cd^{2+} .

The data on closantel (Fig. 5F) is not conclusive since in the authors do not have a positive control: in the control condition presented, the gabazine application did not increase significantly the diffusion coefficient of KCC2. Thus the conclusion on the implication of SPAK/OSR1 in that experiment is unwarranted.

We apologize for this error. The n were obtained from only two experiments and this was not enough to draw firm conclusion. We have now added the results of two new experiments (Figure 5F), showing that gabazine significantly increased (KS test $p < 0.001$) the diffusion of KCC2 in the absence, but not in presence of closantel.

The authors refer to the fact that WNK3 kinase is low in adult neurons (Discussion), but the results shown in Fig 5C-H appear to reveal significant WNK3 expression in hippocampus (albeit 9 fold lower than WNK1). Can the authors discard any significant contribution of WNK3 to the results?

In order to comply with the referee's comment, we have now tested the contribution of WNK3 in the regulation of KCC2 diffusion, clustering and spine head area with a shRNA approach. The results report that WNK3 suppression by RNA interference i) blocks gabazine effect on KCC2 clustering (Figure S5) but ii) does not prevent gabazine-mediated increase in KCC2 diffusion or spine head diameter (Figure S5). Altogether we conclude from these results that mostly WNK1 and, to a lesser extent, WNK3 kinases are involved in the GABAAR-dependent regulation of KCC2 membrane trafficking.

Recent studies (e.g., ref 18) have implicated involvement of WNK1 in pathological conditions which reflect long-term effects. But in the present study, the authors argue for very rapid effects (30 min treatment with gbz). This should be discussed. Are the two phenomena comparable?

We believe the two experimental conditions are too different to be compared (spinal cord pain model vs impact of GABAergic inhibition on hippocampal neurons, molecular regulation vs animal behavior). However, although significance was not reached at the earliest time point, Kahle and coll. (Science Signaling. 2016 9: 421) proposed an effect of the WNK1 pathway "at both early and delayed time points" (see Figure 1D). This however does not preclude that different mechanisms may be at play in the two experimental conditions. See our answer above.

From what I gather, in Fig 6. KCC2 was immunoprecipitated using KCC2 pT906 antibodies and immunoblotted with conventional KCC2 antibodies. Yet the KCC2 pT906 antibodies were raised against a KCC3A epitope that is highly homologous to other KCCs sequence (ref 27). Since other KCCs are expressed in hippocampal neurons it is possible that some of the effects observed reflect altered expression of other KCCs (e.g., KCC3A, which is particularly abundant

in the CNS). These may have competed with KCC2 binding to protein-G-sepharose-coupled KCC2 pT906 antibodies. Additional control are necessary to rule out contribution of other KCCs (e.g., KCC2 pT906/1007 IP; quantitative Immunoblotting for KCC1, 3, and 4).

We have addressed this point by performing IP anti KCC2 pT906/1007 followed by immunoblotting for KCC1, 3 and 4. As shown in Figure S6 immunoprecipitated lysates were equally recognized by an anti KCC3 antibody but not by anti KCC1 or KCC4 antibodies. In addition gabazine application increased phosphorylation of KCC3 T991 and T1048 residues. These data rule out an implication of KCC1 and KCC4 phosphorylation on homologous threonine phosphorylation sites in the regulation by GABAAR-mediated inhibition. However they show that in addition to KCC2, KCC3 also gets phosphorylated upon gabazine application. Given the low contribution of KCCs other than KCC2 to chloride extrusion under isotonic conditions (see Mercado et al JBC 2006, Acton et al J Neurosci 2012) phosphorylation on KCC3 T991/T1048 may have only a minor impact on net chloride transport.

In Fig 6B, the authors must also provide results with Gbz + shWNK1 and closantel to conclusively show that the phosphorylation state of KCC2 T906/T1007 is really WNK1 and SPAK/OSR1-dependent.

Detecting changes in protein phosphorylation in WB upon shRNA suppression of WNK1 expression in hippocampal cultured neurons is impossible, since the yield of lipotransfection is below 1%. However, we have addressed the question of the implication of the WNK/SPAK/OSR signaling pathway in the phosphorylation of KCC2 and NKCC1 by a pharmacological blockade of SPAK activity with closantel. We found that closantel efficiently inhibited the phosphorylation/activation of SPAK/OSR1 but not of the upstream kinase WNK1 (Figure S4). This lack of activation of SPAK/OSR1 in turn prevented the phosphorylation of KCC2 T906/T1007 and NKCC1 T203/T207/T212 (Figure S4).

It is not clear why the western blot samples were heated for 1h at 37{degree sign}C? This is not a conventional approach to remove proteins from beads and denature proteins. The more conventional approach involves heating for ~5 min at ~95{degree sign}C, especially for KCC proteins. The method used here may explain why more oligomers are observed than monomers in Fig 6A and 7E. To show results in denaturing conditions, monomers must be in higher proportions than oligomers. The approach used may thus confound the interpretation of the results.

We followed a published protocol of surface biotinylation using 1h denaturation at 37°C (Mahadevan et al., 2014; Cell Reports 7 (6): 1762-1770). As stated in a review by Medina et al. (Medina et al., 2014; Front Cell Neurosci) “KCC2 is prone to forming SDS-resistant high molecular aggregates during standard protein extraction and solubilization procedures that are normally used for denaturation and dissociation of protein complexes“. The higher quantity of oligomeric than monomeric KCC2 in the blot shown in the figure can therefore be assumed to be an experimental artifact.

In Fig 7E, is the molecular weight ladder properly shown? KCC2 monomers have a MW of ~140 kDa. Monomers in the panel are situated closer to the 100 kDa markers than the 150 kDa ones.

We verified the protein ladder. It is properly shown. The fact that KCC2 monomers are closer to the 100 kDa marker than the 150 kDa is due to the use of a gradient 4-12% gel. Since we stopped running the blot a little bit too early, the 150 kDa marker didn't have the time to migrate far enough into the running gel (closer to the monomeric KCC2 band).

In Fig 7 the authors must provide FLAG surface staining and cell surface biotinylation results with Gbz + shWnk1 and closantel, again to convincingly show that KCC2 membrane clustering and stability are Wnk1 and Spak/OSR1-dependent.

We are now showing that GABAAR-dependent regulation of KCC2 clustering is Wnk1 and Spak/OSR1-dependent in Figure 7B. We have also addressed the contribution of Wnk3 in KCC2 clustering by a shRNA approach (Figure S5). Altogether, our results show the gabazine-induced dispersal of KCC2 clusters is mediated by the Wnk1/Wnk3/Spak/OSR signaling pathway.

Minor points:

Fig 2A and B labeling a bit confusing. I suggest changing the color (e.g. grey) for the "2 μ M GABA + L655,708" condition.

Done.

Fig 5F-G. Why are the results for Median EA not shown here?

This is because of a lack of space but the results are shown in the Supplementary Table 3.

In Fig 6F one cannot see whether the results are significantly different for T906/1007 and T906/1007E (as in the experiment shown in Fig 6E). They appear very different in 6E and 6F.

The boxplots representing KCC2 T906/1007 and T906/1007E in Figure 6E and 6F are exactly the same since they were taken from the same experiments. Therefore, the difference between KCC2 T906/1007 and T906/1007E in Figure 6F is highly significant. We have not highlighted this difference in Figure 6F not to overload the figure since the aim of this figure was to test the impact of gabazine on the diffusion of Threonine mutated KCC2 (i.e. KCC2 T906/1007 vs KCC2 T906/1007+gabazine, KCC2 T906/1007A vs KCC2 T906/1007A+gabazine, or KCC2 T906/1007E vs KCC2 T906/1007E+gabazine). The aim of Figure 6E was on the contrary to test the impact of Threonine mutations on KCC2 diffusion at rest.

Fig 7 B-D and F-G. One could improve readability by using color shading for the Musc condition.

Done.

Page 19 line 2: replace "(Figure 7D)" by "(Figure 7C-D)".

Done.

Reviewer #3 (Remarks to the Author):

In this paper the authors provide evidence for GABA-A receptor mediated regulation of KCC2 abundance and clustering via intracellular Cl⁻ in cultured hippocampal neurons. The data presented and the conclusions drawn are novel and present a model for homeostatic regulation that is interesting and adds to the field.

We wish to thank the reviewer for his/her appreciation of our work.

I have several comments that should be addressed and that could strengthen the project.

Major:

This is entirely an *in vitro* study. The major effect is with GABA-A receptor inhibition rather than activation. Is there evidence for this mechanism being important for endogenous GABA signaling in the intact nervous system?

It is indeed important to know whether the GABAAR-mediated regulation of KCC2 occurs in the intact hippocampal network. We have now addressed this point in Figure 10. First, we showed in hippocampal cultures that the gabazine-mediated regulation of KCC2 clustering also occur in absence of blockers of action potential and glutamate receptors (Figure S8), favoring the hypothesis that this regulation is not only observed under artificial experimental conditions and that it may occur in a more physiological context.

Second, we explored the impact of GABAAR activation in the absence of sodium channel and glutamate receptor blockers on the regulation of KCC2 membrane stability in the intact hippocampal network. For this purpose, we performed surface biotinylation experiments in acute hippocampal slices prepared from 5-7 week old C57bl6 mice treated with vehicle or muscimol for 30 min at 35°C. Although surface biotinylation assays in acute slices are known to be technically challenging due to the high variability between samples, we found that acute muscimol treatment increased by 1.2 and 1.3 fold the surface expression level of KCC2 monomers and oligomers, respectively (Figure S8). Therefore, we concluded that GABAAR activation stabilizes KCC2 at the neuronal surface in the intact hippocampal network.

Finally, we investigated whether GABAAR-dependent KCC2 regulation also occurs *in vivo*. The WNK/SPAK/OSR signaling pathway is virtually silent in the mature brain. We then asked whether this pathway could be “reactivated” in the adult brain under certain circumstances. Since we showed *in vitro* the WNK/SPAK/OSR signaling pathway is activated by lowering GABAergic inhibition, we reasoned this pathway may be activated *in vivo* as well by blocking GABAARs. We therefore tested the effects of a single, subcutaneous injection of the GABAAR antagonist pentylentetrazole (PTZ), a paradigm known to transiently induce epileptic seizures in mice. We now show that the WNK/SPAK/OSR signaling pathway is strongly activated by phosphorylation both in the hippocampus and cortex, 35 min after a single PTZ injection. This effect is associated with an increased phosphorylation of KCC2 and NKCC1 (Figure 10). These data clearly show that KCC2 regulation by the WNK/SPAK/OSR signaling pathway may occur *in vivo* at least under pathological conditions.

We believe these additional data significantly increase the overall impact of our study.

The authors do not report n values for any of the results.

All the ns are now indicated in the figure legends and the Supplementary Tables.

Most comparisons are analyzed with t-tests or Mann-Whitney tests. Data in Figs 6 and 7, however, seem to involve multiple comparisons. For example, T906/1007A and T906/1007E conditions in 6A and B appear to be the same—that being the case why were they not compared for all conditions using a Kuskal-Wallis test? Same is true for WNK1-KD data in Figs. 5 G and H.

In all graphs, we compared a given condition to its control with similar n values. This therefore does not require the use of the non-parametric Kuskal-Wallis test that is used for comparing more than two independent samples of equal or different sample sizes.

Also there were no statistical analyses of the time-averaged MSD results in Figs. 1 and 4.

It is not possible to compare MSD curves with a statistical test. This is why we calculated the explored area that corresponds to the MSD value of each trajectory at two different time intervals of 0.42 and 0.45 s (see Materials and Methods). Therefore, the statistical analyses of the time-averaged MSD in Figs. 1C, 1G, 4C, 4G are shown in Figs. 1D, 1H, 4D, 4H.

The shRNA experiment lacks a control for the vector (with an irrelevant shRNA sequence) and either a second shRNA or a cDNA rescue to control for off-target effects.

We are sorry for the error in Figure 5H. The control of the WNK1 shRNA experiment is in fact a non-target shRNA. The on-target sequence was replaced by a mock sequence of the WNK1 shRNA in the same vector backbone. Furthermore, we now have expressed a second non-target shRNA that was expressed in a distinct vector. Neurons expressing this non-target shRNA showed gabazine-dependent increase in KCC2 diffusion similar to neurons expressing the shMock (Figure 5H, Figure S5C).

Central to the model is the modulation of $[Cl^-]_i$, which was indirectly measured by a depolarizing shift in transfected glycine receptor reversal potential induced by gabazine. A direct measure using a fluorescent indicator would strengthen the argument, as well as testing some other pharmacological manipulations in which $[Cl^-]_i$ was assumed to be altered, such as muscimol, closantel, and VU 0240551

We have now directly addressed this point by FRET imaging of the chloride sensor SuperClomeleon in hippocampal neurons (see reply to referee 2 above).

Minor:

In many places the words "compared to" are used but the comparison described differences in two conditions that are similar. When describing differences between otherwise

We have double-checked the manuscript and we believe "compared to" was used appropriately as it refers to comparison of one experimental condition versus a control condition.

REVIEWERS' COMMENTS:

Reviewer #1 (Remarks to the Author):

The authors have adequately answered all issues raised by their initial submission.

Reviewer #3 (Remarks to the Author):

The additional experiments have improved the paper and answered most of my concerns. The addition of chloride sensor measures is a plus as are the new slice and neuron experiments. The request for a second shRNA or a rescue experiment for WNK1 knockdown was not addressed. However, given that the kinase dead mutant over expression gave the same result, I do not think this is essential. The authors should be aware though that off target effects of single shRNAs are quite common and results should be validated with a rescue cDNA or replicated with a second shRNA that targets a different sequence in the gene of interest (or in this case with another approach such as the KD mutant).

With regards to the use of "compared to" versus "compared with" I would draw the authors' attention to the discussion of usage in "Elements of Style" by Strunk and White .

Please find below our response to reviewers' comments.

Reviewer #1 (Remarks to the Author):

The authors have adequately answered all issues raised by their initial submission.

We are thankful to the reviewer for his/her recognition of the accomplished work.

Reviewer #3 (Remarks to the Author):

The additional experiments have improved the paper and answered most of my concerns. The addition of chloride sensor measures is a plus as are the new slice and neuron experiments.

We are thankful to the reviewer for his/her enthusiasm and recognition of the accomplished work.

The request for a second shRNA or a rescue experiment for WNK1 knockdown was not addressed. However, given that the kinase dead mutant over expression gave the same result, I do not think this is essential. The authors should be aware though that off target effects of single shRNAs are quite common and results should be validated with a rescue cDNA or replicated with a second shRNA that targets a different sequence in the gene of interest (or in this case with another approach such as the KD mutant).

We agree with the reviewer that single shRNAs may have off targets effects. Two different WNK3 shRNAs were raised in the lab and used to prevent off targets effects of single shRNAs. Concerning WNK1 shRNA, in absence of a second shRNA to suppress WNK1 or of a rescue construct, we used the kinase dead mutant overexpression strategy. Both the shRNA and the kinase dead mutant gave the same results: they blocked the gabazine effects on KCC2 membrane diffusion and clustering as well as on spine head growth. These results were further reproduced by the chemical inhibition of WNK1 cascade and the use of KCC2 T906/T1007A and T906/T1007E mutants, the KCC2 residues targeted by WNK1. Furthermore, the shRNA against WNK1 was characterized and used previously (for a recent citation see Science Signaling 30 June 2015 Vol 8 Issue 383 ra65). In this publication, a similar shRNA/kinase dead replacement strategy was used. Altogether, we believe our data indicate the WNK1 shRNA effects were not due to off target effects of the shRNA. We therefore think it is not essential to use a second shRNA or a rescue construct. This also appears to be the opinion of the reviewer: **“given that the kinase dead mutant over expression gave the same result, I do not think this is essential.”**

With regards to the use of "compared to" versus "compared with" I would draw the authors' attention to the discussion of usage in "Elements of Style" by Strunk and White.

The text was modified as recommended by the reviewer.